**Article** https://doi.org/10.1038/s41467-023-42898-9

# *replicAnt*: a pipeline for generating annotated images of animals in complex environments using Unreal Engine

Fabian Plum [1] ✉, René Bulla [2], Hendrik K. Beck [1], Natalie Imirzian[1] & David Labonte [1] ✉

Deep learning-based computer vision methods are transforming animal behavioural research. Transfer learning has enabled work in non-model species, but still requires hand-annotation of example footage, and is only performant in well-defined conditions. To help overcome these limitations, we developed *replicAnt*, a configurable pipeline implemented in Unreal Engine 5 and Python, designed to generate large and variable training datasets on consumer-grade hardware. *replicAnt* places 3D animal models into complex, procedurally generated environments, from which automatically annotated images can be exported. We demonstrate that synthetic data generated with *replicAnt* can significantly reduce the hand-annotation required to achieve benchmark performance in common applications such as animal detection, tracking, pose-estimation, and semantic segmentation. We also show that it increases the subject-specificity and domain-invariance of the trained networks, thereby conferring robustness. In some applications, *replicAnt* may even remove the need for hand-annotation altogether. It thus represents a significant step towards porting deep learning-based computer vision tools to the field.

Enabled by the continued reduction in cost of computational hardware and breakthroughs in deep neural network architectures and training paradigms, data-driven deep learning approaches now represent the state of the art in almost all computer vision applications[1,2]. This success has been achieved in discriminative applications such as classification[3], detection[4], pose-estimation[5], and semantic segmentation[6], as much as in generative applications, as demonstrated by recent advancements in diffusion networks which can create stylised and near photo-realistic images from text prompts[7,8]. Both discriminative and generative approaches have in common that they primarily involve supervised learning, which, to an extent, resembles high dimensional interpolation: achieving generalisability is practically synonymous with ensuring that inputs at training time reasonably resemble those encountered at inference time. As an illustrative example, successful detection requires that instances of the target class are identified regardless of image context and subject appearance[4]; the ideal detector is subject-specific, but domain-invariant. Large, curated and annotated datasets—such as those provided by ImageNet[9], COCO[10], or CiFAR[11]—are indispensable in this process, as they provide a basis for learnable real-world principles, and complex testing grounds.

A prime area of application for the emerging machine learning toolset is animal behavioural research[12–18], where it promises to reduce time costs, increase statistical power, and minimise potential for human bias; machine learning may altogether revolutionise what is possible in ethology[13,18], and its intersection with neuroscience[16,19,20], morphology[21], locomotion[13,18,22], and conservation[23]. Despite the divergence in the questions tackled, all applications in these research areas have in common the need for annotated training data. Unfortunately, datasets of a size and quality required to achieve robust

---

[1]Department of Bioengineering, Imperial College London, London, UK. [2]The Pocket Dimension, Muenchen, Germany.
✉e-mail: fabian.plum18@imperial.ac.uk; d.labonte@imperial.ac.uk

domain-invariant inference are rarely available, and—apart from a few model species such as mice or *Drosophila*—the effort required to curate them often outweighs the immediate benefit of the enabled automation. Transfer learning—i.e., pre-training (parts of) a network on a separate, much larger, dataset, and refining the network on a small number of hand-annotated images—is a strategy that has been implemented with great success in markerless animal pose-estimation[16,17,20]. However, the price paid for the substantial reduction in the necessary amount of hand-annotation is that the resulting networks are typically only performant under stereotyped conditions, and frequently require extensive input pre-processing. Even minute deviations from the refinement data—for example in form of partial occlusion or changes in specimen appearance, lighting, background, perspective, or camera type—can result in a substantial drop in network performance. As a result, transfer-learning strategies perform best in well-controlled recording conditions, and additional refinement is required to analyse more variable footage from the gold standard of behavioural studies— field experiments. Although refinement with relatively few hand-annotated samples of the order of a few hundred to a few thousand can enable accurate inference under field conditions[13,17,20,23–26], large appearance deviations from the hand-annotated examples—for example, due to changes in weather conditions, recording background, the time of day, or varying camera perspective—typically considerably decrease performance[17,19,23,27]: networks learn latent features specific to the recording environment, rather than a general subject-specific understanding. Some of these generalisation issues can be addressed through data augmentation, i.e. the application of image perturbations with the aim to alter image appearance while retaining its meaning and label[4,28,29]. For example, by changing the rotation, scale, hue, and resolution of an image, its contents would still remain identifiable. More sophisticated augmentation strategies, such as style transfer, can further improve network robustness[30–32]. Alternatively, where large volumes of unlabelled data are available, self-supervised approaches may be employed to learn consistently identifiable features[24,33]. But these features may then be distinct from case-specific points of interests, in some sense just passing the baton of key-point extraction further down in the analysis pipeline. Currently, even extensive augmentation and unsupervised or self-supervised strategies still pale in their efficacy in comparison to simply using larger and more varied datasets in supervised approaches instead[4,28,29].

In robotic[34–36], human[37–40], and automated driving[41–43] applications, annotated datasets comprising billions of images can now be produced "synthetically", i.e. through simulation with a computer. By placing 3D models in simulated environments, variable and annotated datasets can be generated at scale, and at a fraction of the cost and time required for hand-annotation of real images[39,40,42,44]. The use of synthetic data is particularly attractive where annotated real datasets are practically absent or only of insufficient size, as is the case for almost all non-human animal studies[22,30–32,45–50]. However, for all its conceptual attractiveness, using synthetic data is not without problems: the simulated images must bridge the "simulation-reality gap", i.e. they must be comparable in appearance to real images; as before, the key challenge remains that the training data must represent a superset of the inputs received at inference time[22,27,47–49]. As an illustrative example, Arent et al.[22] modelled Indian stick insects as a rigid body consisting of simple geometrical shapes to improve the performance of a DeepLabCut[20] pose estimator. Such simplified geometric approaches can improve performance, but remain restricted to stereotyped recording settings, simple animal morphology, and a single output data type. More complex approaches have used hand-animated or learned motion priors, or combined low fidelity synthetic data with style- or domain transfer networks to close the simulation-reality gap[27,30,31,47–50]. These approaches however remain labour-intense, tied to specific species, possess limited options for annotations, or still require extensive real image datasets in order to generalise to real

examples. Comprehensive and generalisable approaches which utilise more realistic animal representations, handle large digital animal populations, can create highly variable environments, and provide options for complex annotation, remain absent.

Here, we address this gap and present a synthetic dataset generator, *replicAnt*, implemented in Unreal Engine 5, a 3D computer graphics game engine, and Python. *replicAnt* can be used to simulate the appearance of animals in complex, procedurally generated environments with all but a few clicks of a mouse. Leveraging recent advancements in photogrammetry, real-time ray tracing, and high-resolution mesh handling, *replicAnt* runs on consumer-grade computational hardware, automatically produces rich image annotations, and can simulate virtually any recording conditions, including variations in camera model and perspective; individual number, size, pose, and colouration; scene lighting; image resolution and magnification; and environment appearance. We demonstrate the versatility and utility of *replicAnt* by using the synthetic data it generates to train deep neural networks for automatic inference in four common animal applications: (1) detection—localising animals in an image; (2) tracking—retaining the identity of animal detections across continuous frames; (3) markerless pose-estimation—extracting the coordinates of user-defined body landmarks; and (4) semantic and instance segmentation—determining which areas of an image correspond to an animal on a pixel level.

## Results
### replicAnt
*replicAnt* uses 3D models of animals to produce a user-defined number of annotated images. It is designed to generate large and variable datasets involving hundreds of animals with minimal user effort; due to the rich and automated annotation, a single synthetic dataset can then be used to train a variety of deep neural networks. *replicAnt* requires: 3D models of the study organism(s); the installation of a pre-configured Unreal Engine project; and custom-written data parsers, used to translate the generated data into formats compatible with the deep learning-based computer vision system(s) of choice (see Fig. 1).

*replicAnt* is agnostic to the origin of the subject 3D model(s) used as input. Throughout this work, we use high resolution 3D models produced with the open-source photogrammetry platform *scAnt* (Fig. 1a)[51]; but we also demonstrate that simpler hand-sculpted models can suffice for some applications. In general, the higher the 3D model fidelity, the higher the application flexibility.

Depending on their origin, 3D models may need to be cleaned, and—if the randomised pose variation feature of *replicAnt* is to be used —virtual bones and joints need to be assigned, and their range of movement defined (see Fig. 1c, "Methods—3D subject models", and the *replicAnt* GitHub https://github.com/evo-biomech/replicAnt). In this paper, we focus on insects, first because of personal predilection, and second because an exoskeleton avoids the need to simulate the complex soft tissue deformation associated with postural changes in animals with endoskeletons. However, powerful approaches to create photo-realistic models of vertebrates exist[52–54], and *replicAnt* is not limited to arthropod models (or even just animals, for what it is worth). The cleaned and rigged model is imported into a pre-configured Unreal Engine 5 project, where a simplified collision mesh is computed to enable interactions with objects inside the simulated world (Fig. 1d).

Next, a customisable digital "population" is generated by simulating multiple instances of the original subject model. Variation between subject instances is achieved through simple appearance modifications, such as changes in brightness, contrast, hue, saturation, and scale. The range of these modulations can be adjusted through a simple user interface, and custom modifications can be added. Subjects are later sampled at random from this population, and placed into procedurally generated environments, from which annotated images are extracted.

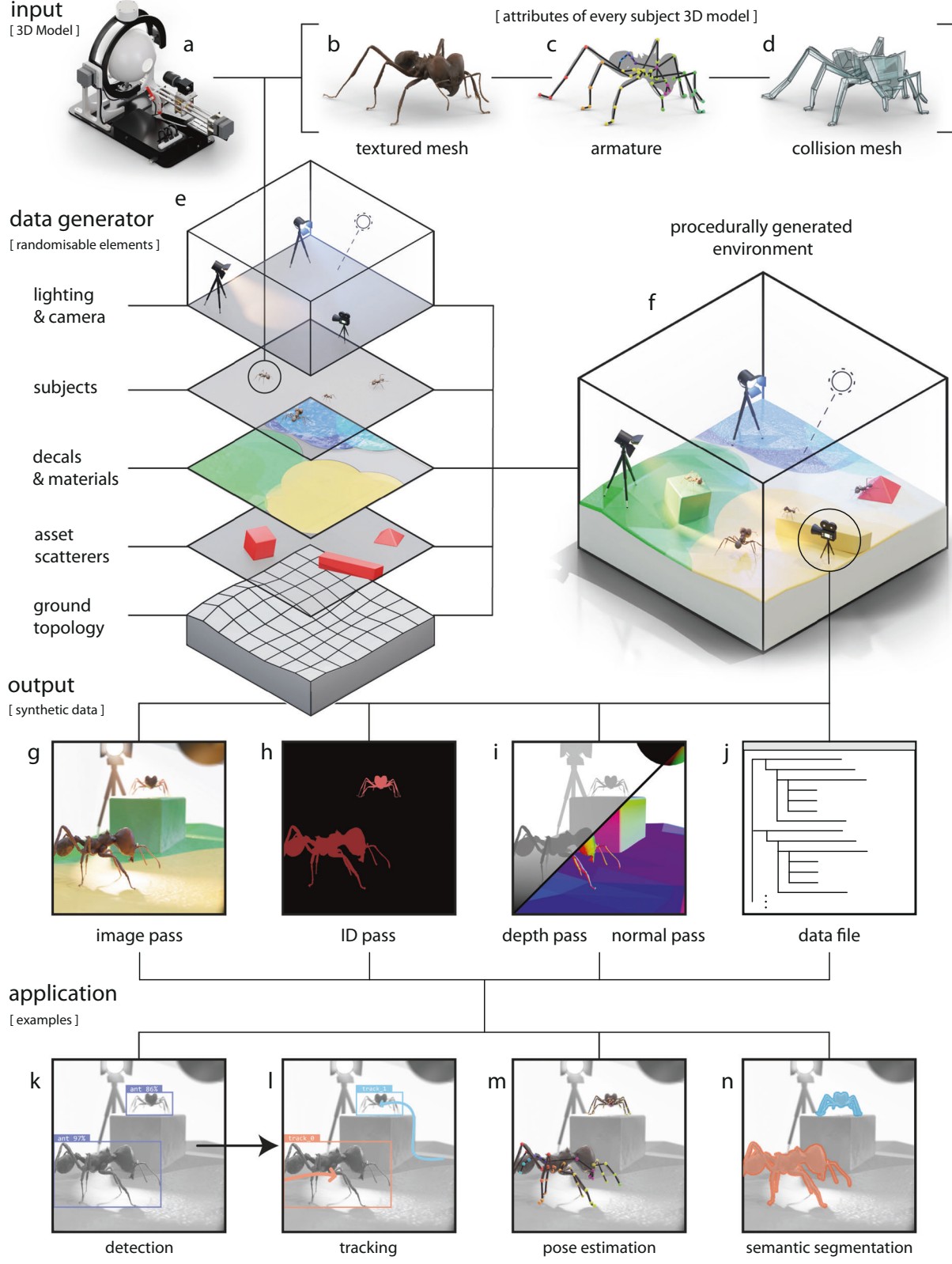

Each scene is generated in a hierarchical process, structured into five customisable levels to maximise computational efficiency; changes in lower-level hierarchical elements influence higher level elements (Fig. 1e). At the lowest level of scene hierarchy sits a ground plane with random topology. At the second level, this ground plane is populated with 3D assets; polygon meshes of objects such as plants, rocks, and common household items, all of random size. Assets are drawn from a curated library, and placed by a configurable number of asset scatterers. At the third level, the ground and each asset are assigned Physically Based Rendering materials, generated by blending randomly generated patterns with a curated texture library. Large material maps, or decals, are generated and wrapped around the ground plane and all assets to achieve a cohesive scene appearance. At the fourth level, a configurable number of subjects from the model

**Fig. 1 | *replicAnt* is a toolbox designed to procedurally generate and automatically annotate image samples from 3D animal models.** The combination of images and annotations constitutes "synthetic data" which can be used in a wide range of deep learning-based computer vision applications. **a** *replicAnt* requires digital 3D subject models; all but one subject model used in this work were generated with the open-source photogrammetry platform *scAnt*[51]. Each model comprises **b** a textured mesh, **c** an armature, defined by virtual bones and joints, to provide control over animal pose, and **d** a low-polygonal collision mesh to enable interaction of the model with objects in its environment. **e** 3D models are placed within environments procedurally generated with a pre-configured yet customisable project in Unreal Engine 5. **f** Every scene consists of the same core elements, configurable via dedicated randomisation routines to maximise variability in the generated data. 3D assets are scattered on a ground of varying topology; layered materials, decals, and light sources introduce further sources of variability across scene iterations (see examples in Figs. 2–6). From each scene, we generate **g** image, **h** ID, **i** depth, and normal passes, accompanied by **j** a human-readable data file which contains annotations and key information on image content (see "Methods" for details). Synthetic datasets generated with *replicAnt* can then be parsed to train networks for a wide range of computer vision applications in animal behavioural research, including **k** detection, **l** tracking, **m** 2D and 3D pose-estimation, and **n** semantic segmentation.

population are placed at random locations, and their pose is adjusted via inverse kinematics, such that they can interact with the surrounding meshes. At the fifth and highest hierarchical level, scene lighting is introduced in form of a configurable number of coloured light sources and High Dynamic Range Images (HDRIs), and a virtual camera with randomisable extrinsics, intrinsics and post processing parameters is placed; the scene generation is now complete (Fig. 1f).

Using the virtual camera, "image passes" are exported from each scene iteration. Each pass encodes different information (Fig. 1g–j), for example the optical image render itself, or depth information (for details, see "Methods"). User-defined passes can be added as required. Each image pass set is supplemented by a data file which contains configurable annotations, for example subject bounding boxes, 2D and 3D key point coordinates, class labels, or camera intrinsics and extrinsics. The combination of image passes and data files constitute synthetic data which can be used to train deep neural networks for various computer vision tasks (Fig. 1k–n).

The entire process, including the generation of a user-specified number of scene iterations, image pass rendering, and data file writing, is fully automated, but leaves open plenty of opportunity to introduce variation with minimal effort. The pre-configured Unreal project, detailed documentation, and additional resources are available from https://github.com/evo-biomech/replicAnt.

## Applications

In order to demonstrate that the synthetic data generated by *replicAnt* is of sufficient quality to power applications in animal behavioural research, we used it to train various popular deep learning networks for animal detection, tracking, pose-estimation, and semantic and instance segmentation. The performance of these networks was then evaluated on dedicated example datasets. Unless stated otherwise, all synthetic data used for training was generated using *replicAnt*'s default settings (see "Methods−Data parsers" and GitHub for details). We will now show that *replicAnt* significantly improves the trained networks' ability to generalise to unseen conditions; in some cases, it removes the need for hand-annotation altogether, and in others, it may present the only option to generate datasets large enough to train robust and performant networks in reasonable time.

**Detection.** A digital population of *Atta vollenweideri* leafcutter ants (Forel 1893), comprising 100 simulated individuals, was created using 3D models of a minor, media, and major worker, all generated with *scAnt*[51] (Fig. 2b, see "Methods−Detection" for details). This population formed the basis for two synthetic datasets, each encompassing 10,000 annotated images with a resolution of 1024 × 1024 px: one used all three 3D models ("group"), and one using only the largest model ("single"). Furthermore, to investigate the influence of synthetic dataset size on inference performance, networks were trained on 1% ("small"), 10% ("medium"), and 100% ("large") of the "group" dataset. Dataset generation took about ten hours each for the full "group" and "single" datasets on a consumer-grade laptop (6 Core 4 GHz CPU, 16 GB RAM, RTX 2070 Super).

The generated synthetic datasets were then used to train a commonly used object detector, YOLOv4[4], subsequently tested on laboratory recordings of a crowded foraging trail (Fig. 2d). Foraging trails of *Atta* ants present an ideal example for complex detection tasks as individuals vary in size, trails are highly cluttered, and partial as well as full occlusions occur frequently. In order to introduce variation in scene appearance, akin to what may be expected in field conditions, scene lighting, exposure time, camera magnification and foraging trail background were altered systemically, yielding five different recording scenarios (Fig. 2e). For each recording scenario, 1000 frames each with between 36 to 103 individuals were hand-annotated using BlenderMotionExport[55]. For comparison, we also trained detectors on 5000 of these hand-annotated images, using image combinations from the different recording scenarios. Five-fold cross validation, with 80/20 splits between training and validation data, was used for all training (see "Methods" for details on test data and training schedules).

In general, detectors performed best on within-domain data, where they achieved close to perfect performance (Fig. 2g). The notable exception to this rule were close-up recordings, where the detector trained on synthetic data outperformed the within-domain network. However, the performance of detectors trained with real data dropped notably when they were used for inference on unseen images, despite the similarity in perspective (Fig. 2g, h). In sharp contrast, the detectors trained with synthetic data retained a robust and consistent performance throughout (Fig. 2g). To quantify this difference, the Average Precision (AP) was averaged, so yielding a mean Average Precision across all unseen test cases (mAP, see Eq. (2)). The detectors trained exclusively with synthetic data achieved an mAP of $0.913 \pm 0.0079$, both higher and less variable than all detectors trained with any of the five real sub-datasets (Fig. 2g). For comparison, the best real data detector was trained on noisy images, and achieved an mAP of $0.878 \pm 0.0258$. Networks trained exclusively on synthetic data converged more slowly and exhibited an overall higher loss during training compared to any set of real training images, indicating a higher level of complexity of the generated images (Fig. 2f). These results indicate that the large volume and variability of synthetic data substantially increases robustness of detections; supplementing training datasets with synthetically generated samples may be a suitable strategy to significantly reduce the hand-annotation required to achieve benchmark performance, and can improve the ability of networks to generalise to unseen conditions. To test these ideas, detectors were trained on "mixed" datasets, containing both real and synthetically generated images (see methods for details). Networks trained with a 10,000/100 synthetic/real split ("sb1") achieved an mAP of $0.9501 \pm 0.014$, close to the benchmark performance (Fig. 2g). A more extensive quantitative comparison of the performance across inference cases is provided in the Supplementary Table 5. In order to confirm that synthetic data enables networks to recognise ants specifically and not just objects with similar appearance, we tested detectors trained with 3D models of desert termites (*Gnathamitermes* sp., see below), which resulted in a negligible mAP of $0.007 \pm 0.005$ (Fig. 2g).

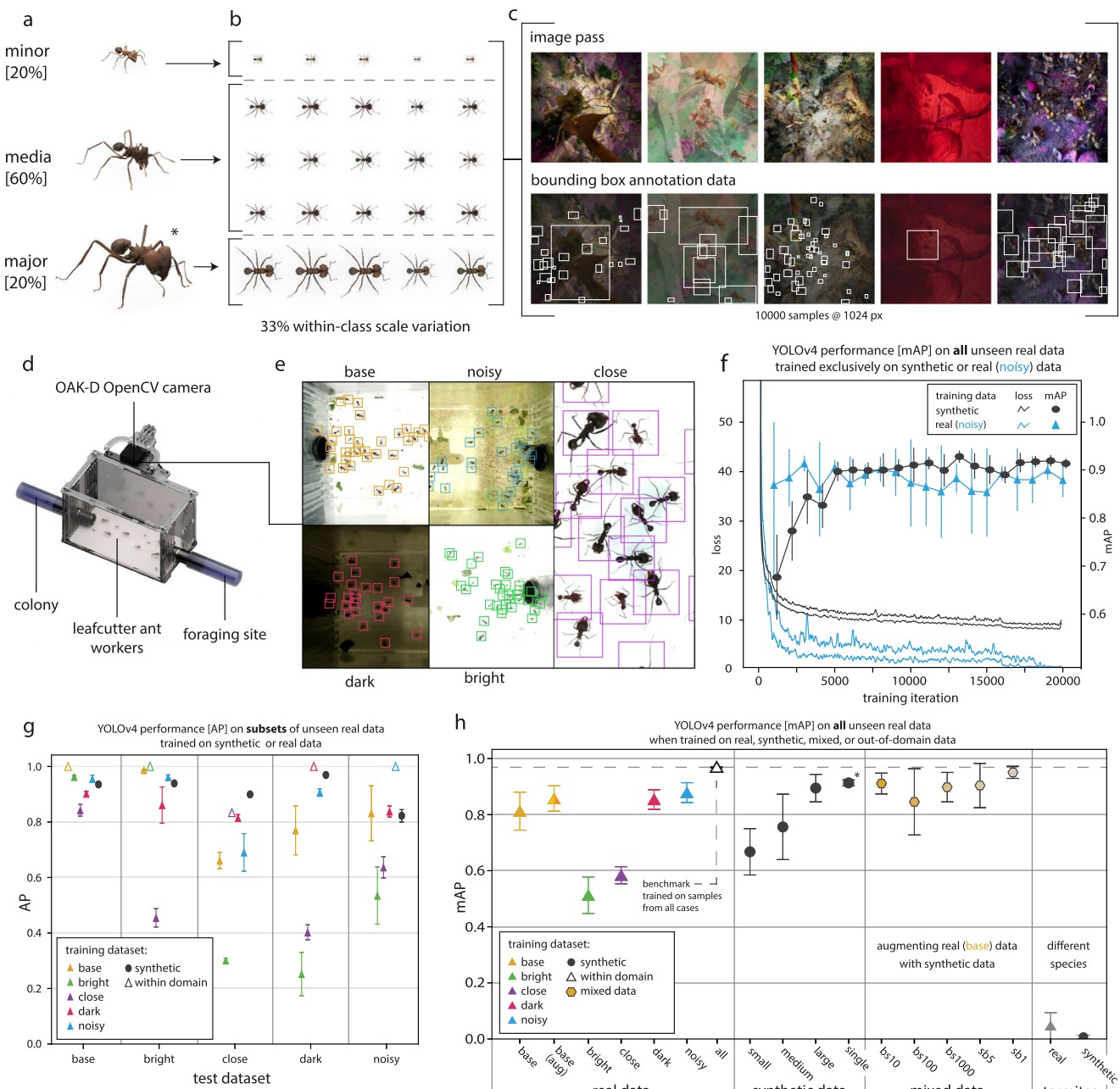

**Fig. 2 | Performance of YOLOv4 detectors trained with real, synthetic and "mixed" data. a** 3D models of leafcutter ant workers, created with *scAnt*[51], form the basis of **b** a digital population comprising 100 individuals which differ in scale, hue, contrast, and saturation. *replicAnt* was then used to produce synthetic datasets with 10,000 annotated samples from this population. **c** Examples of image render passes (top row) and bounding box annotations (bottom row). **d** Test data were obtained with a laboratory setup consisting of an OAK-D OpenCV camera, which recorded foraging trails of *Atta* leafcutter ants from a top-down perspective. **e** Recording conditions were varied to produce five sub-datasets of varying difficulty (see "Methods" for details). **f** YOLOv4[4] networks converged more slowly when trained on synthetic data, indicating a more complex training taskbut ultimately yielded higher mean Average Precision (mAP) scores. **g** Networks trained on real images from any sub-dataset perform poorly on out-of-domain recordings, as indicated by the low average precision (AP) of the detections (solid triangles). In notable contrast, networks trained solely on synthetic data achieved a strong detection performance throughout, likely because they have been exposed to considerably larger variation at training time (black circles). **h** The superior performance of networks trained on synthetic data is most apparent when the mAP is compared directly: The highest mAP of $0.951 \pm 0.014$ was achieved by a network trained on a mix of synthetic and real data (case sb1, 10,000 synthetic and 100 real images; see "Methods" and Supplementary Tables 1–6 for a full breakdown of each dataset including sample sizes). Remarkably, the second highest and most consistent mAP was achieved by a network trained solely on synthetic data ($0.913 \pm 0.001$, marked with an asterisk). Error bars (**f–h**) indicate the standard deviation of the respective mean with fivefold cross-validation using different withheld data splits during training. Source data are provided as a Source data file.

Next, we sought to demonstrate that high model fidelity is not required for detection tasks which typically involve low magnification recordings. Instead, even simpler hand-sculptured models can be used to train performant networks, powered by the large volume and variability of training images that can be generated with *replicAnt*. We procured a test dataset of 1000 consecutive frames of 49 freely moving desert termites, *Gnathamitermes* sp., recorded in the field and hand-annotated using BlenderMotionExport[55] to provide a simple benchmark (Fig. 3 and "Methods" for details). Two 3D models, one of a worker and one of a soldier, were hand-sculpted from reference

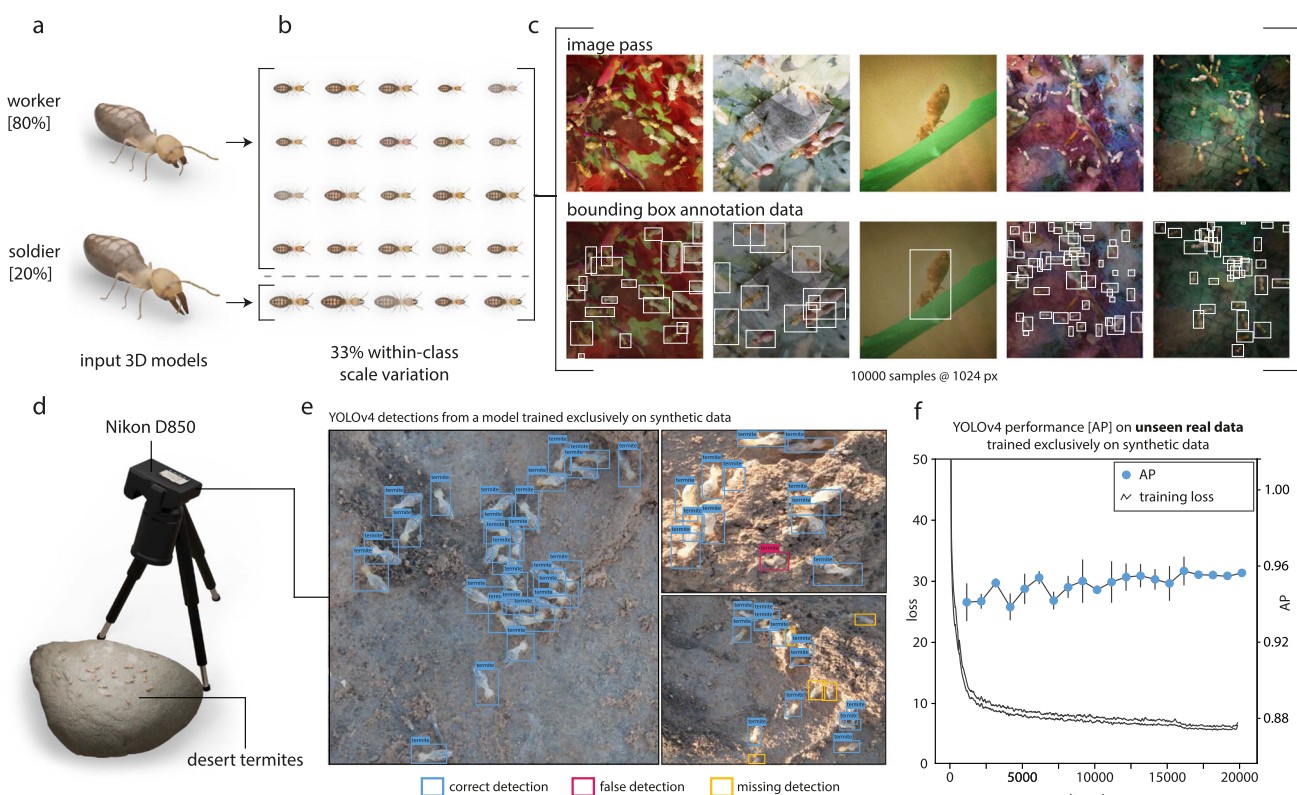

**Fig. 3 | Even low fidelity 3D models can be used to train performant networks for low-magnification applications such as animal detection. a** 3D models of a worker and a soldier desert termite (*Gnathamitermes* sp.) were sculpted and textured from reference images in Blender v3.1. **b** A digital population comprising 80 workers and 20 soldiers, each with randomised scale, hue, contrast, and saturation, was generated from these models (see Supplementary Table 4), and used within *replicAnt* to generate a synthetic dataset with 10,000 annotated images. **c** Examples of image render passes (top row), and bounding box annotations (bottom row). **d** Test data was recorded in the field, using a Nikon D850 and a Nikkor 18–105 mm lens. **e** Example frames demonstrate the high precision of a YOLOv4 detector trained exclusively on synthetic data; only a small number of occluded or blurry individuals were missed, and few false positives were produced (confidence threshold of 0.65, and non-maximum suppression of 0.45). **f** The YOLOv4[4] network converged after about 20000 iterations, and achieved an Average Precision (AP) of $0.956 \pm 0.001$, retrieved from returned detections on 1000 hand-annotated frames of 49 freely moving termites (see "Methods"). Error bars (**f**) indicate the standard deviation of the respective mean AP, computed every 1000 training iterations on the unseen real data with fivefold cross-validation using different withheld synthetic data splits during training. Source data are provided as a Source data file.

images using Blender v3.1. A YOLOv4[4] network, trained on a dataset of 10,000 synthetically generated images with a resolution of $1024 \times 1024$ px (Fig. 3a–c, see Supplementary Table 4 for details), achieved an AP of $0.956 \pm 0.001$ on the annotated recordings, and produced accurate detections in qualitative test cases (Fig. 3e, f).

**Multi-animal tracking.** Sufficiently precise detectors can in principle be used to build simple, yet robust and performant trackers. To facilitate the use of *replicAnt*-trained detectors in tracking applications, we introduce *OmniTrax*[56], an open-source Blender add-on. *OmniTrax* allows users to conduct interactive detection-based buffer-and-recover tracking using imported YOLO detector networks[4,57], and multi-animal pose-estimation, using DeepLabCut (see below, and ref. 20); it also provides extensive annotation options (see Supplementary Videos 1–4 and 7). Tracking is achieved by linking YOLO detections across frames via Kalman-Filtering and the Hungarian method for track association[58]. To assess the performance of this simple tracking architecture, we imported the best performing detection networks trained exclusively on synthetic data into *OmniTrax*, and tracked laboratory and field recordings of *A. vollenweideri* leafcutter ants and *Gnathamitermes* sp. desert termites (Fig. 2). The ant detector tracked between 61 and 103 *A. vollenweideri* ants over 1000 frames at 30 fps, equivalent to real time inference on a consumer-grade laptop (6 core CPU, 16 GB Ram, RTX 2070); the desert termite detector tracked 49 individuals across 1000 frames. Default tracker settings were used for both test cases.

The ant tracker achieved Multiple Object Tracking Accuracy (MOTA) scores of 0.901, 0.945, 0.859, 0.821 in the "base", "dark", "bright", "noisy" cases, respectively (see Eq. (3), ref. 59, and Fig. 4). Most ID switches were caused by track fragmentation, which can be avoided by refining tracker settings, or through simple manual corrections within *OmniTrax*. An extensive quantitative comparison of the performance across inference cases is provided in Supplementary Table 5. The desert termite tracker achieved a MOTA of 0.96. Only two true ID switches occurred; the remaining errors reflect partially fragmented tracks, and a single out-of-focus animal which was not registered (Fig. 4). Thus, and despite its structural simplicity compared to other recent approaches[16,60–62], the detection-based tracker powered by *replicAnt* and implemented in *OmniTrax* can track a large number of animals in crowded and open scenes–without the need to hand-annotate a single image.

**Pose-estimation.** Animal pose-estimation typically leverages transfer-learning. Although excellent performance is possible in controlled settings, the characteristically small training datasets usually fail to provide the variability required for generalisation to unseen recording settings. As a result, performance is extremely sensitive to changes in scene or specimen appearance[15–17]. The key problem is that maximising performance through overfitting of perspective-dependent latent features leads to domain-dependence. Our aim is to overcome this limitation by leveraging the large and variable synthetic datasets produced by *replicAnt* to embed an improved subject understanding into

the pose-estimation networks. In other words, we ultimately seek to train a single generalist network, rather than several scene- and perspective-dependent specialists, as is currently best practice[16,17,20].

To move towards this aim, we used a 3D model of a sunny stick insect (*Sungaya inexpectata* Zompro 1996, first instar) to generate 10

sub-datasets with different randomisation seeds, characterised by 70% scale variation, and hue, brightness, contrast, and saturation shifts producing 1000 samples each. These datasets were combined into one single-animal synthetic dataset encompassing 10,000 images at a resolution of 1500 × 1500 px; the automated annotations included the

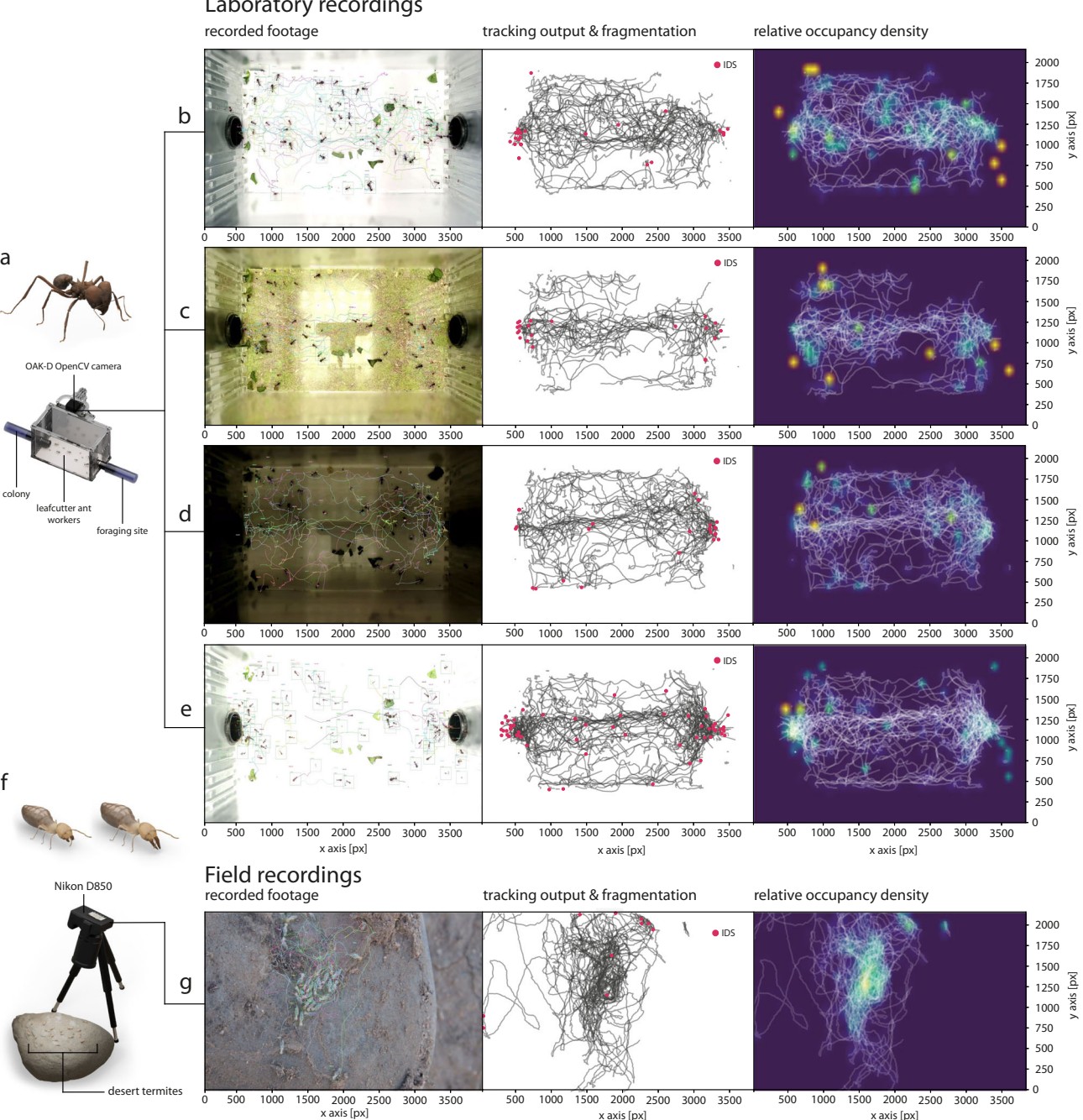

**Fig. 4 | Detectors trained exclusively on synthetic data can be used for multi-animal buffer-and-recover tracking. a** A YOLOv4 network[4] was trained on synthetic images generated from a single *Atta vollenweideri* specimen, and then used within *OmniTrax*[56] to automatically track individual ants in **b** well-exposed, **c** noisy, **d** dark and **e** bright (over-exposed) footage (left column). Detections are provided by the YOLOv4 network[4], and linked across frames through a simple buffer-and-recover approach, using a 2D Kalman-Filter implementation and the Hungarian method for track association cost assignment[58]. **b**–**e** Tracking performance was evaluated by tracking between 61 and 103 individuals, which could freely enter and exit the recording area, over 1000 frames, with up to 62 animals present simultaneously. ID switches (IDS) are marked in red and mostly occur at the entrances to

the recording site. They typically result from track fragmentation due to prolonged occlusion, and can thus be easily excluded from further analysis (middle column). Tracking performance deteriorates with overexposed images (**e**), fuelled by a combination of motion blur and drastic changes in appearance due to exposure clipping. Relative occupancy density maps visualise cluttered areas, path preferences, and static individuals (right column). **f**, **g** A YOLOv4 network[4], trained exclusively on synthetic images of desert termites (Gnathamitermes sp.) was used to track 49 desert termites across 1000 frames from field recordings. Only two true identity switches occurred (IDS, entrance at the top right corner); all other IDS are the result of fragmented tracks. Source data are provided as a Source data file.

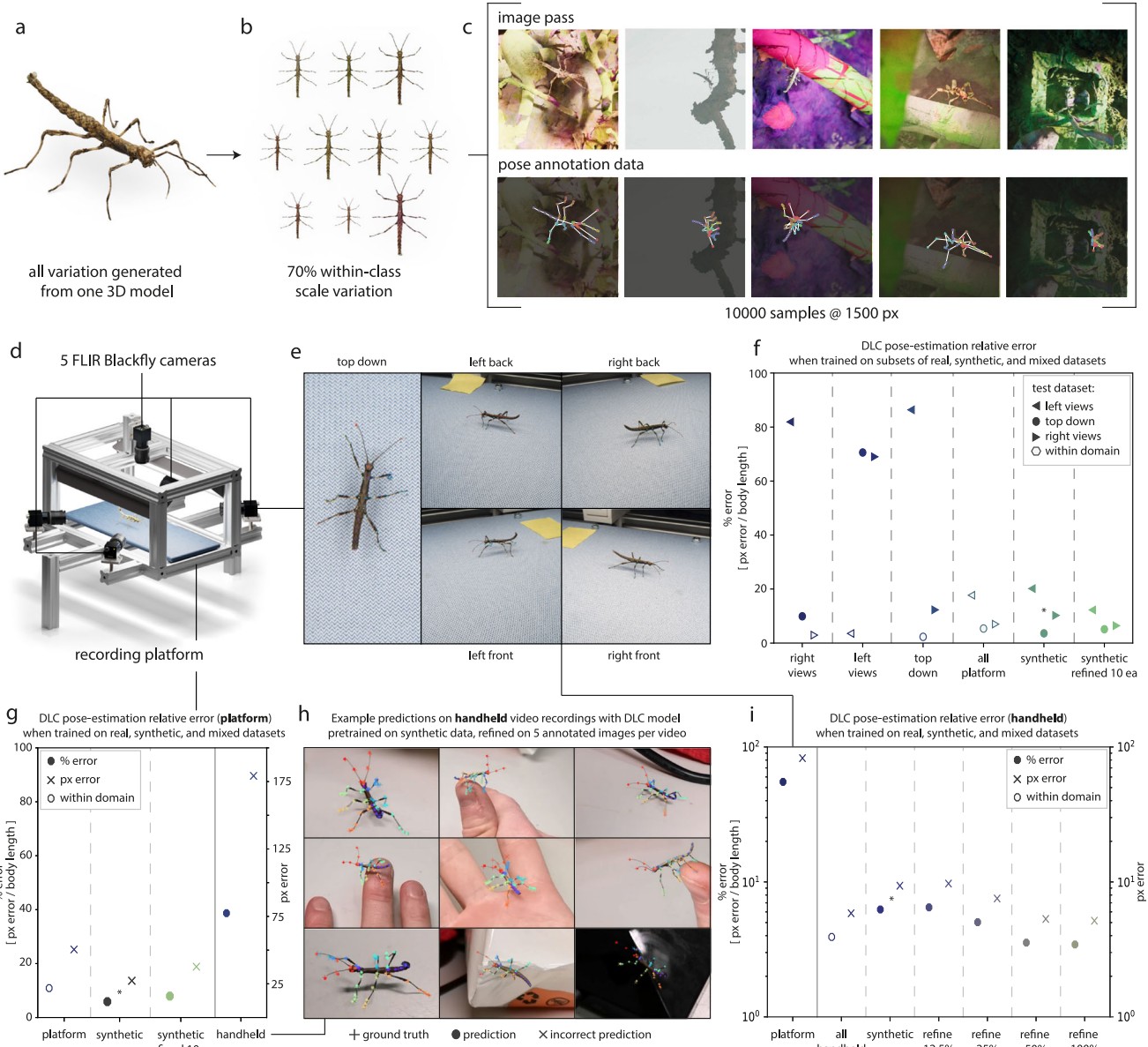

**Fig. 5 | Performance of DeepLabCut (DLC) markerless pose-estimators[20] trained on real, synthetic and mixed datasets. a, b** A digital population of 10 individuals was generated from a single 3D model of a *Sungaya inexpectata* stick insect, created with the open-source photogrammetry platform *scAnt*[51]. Population subjects differed in scale, hue, contrast, and saturation, and formed the basis for a synthetic dataset. **c** Examples of image render passes and key point annotation; the locations of the 46 key points are provided in Supplementary Fig. 1. A DeepLabCut[20] network with a ResNet101 backbone was then pre-trained on the synthetic dataset for 800k iterations, and fine-tuned on splits of real samples for a further 800k iterations (see "Methods" and Supplementary Tables 8–11 for details). Two hand-annotated datasets serve as test cases: **d, e** five synchronised machine vision cameras mounted to a tiltable platform recorded walking stick insects in controlled conditions, so that only camera perspective varied; **h** handheld cell phone recordings of walking stick insects across the laboratory, which include partial occlusions, and variations in background and lighting. **f, g** Networks pre-trained on synthetic data and refined

with just 10 samples per camera perspective achieved a mean relative error of 8.14%, and a mean pixel error of 37.69 px across orientations—lower than the benchmark performance achieved by the network trained on the full dataset of 805 total example images (mean relative error of 10.90%, mean pixel error of 50.57 px). The asterisk, *, indicates networks for which ≤50% of the inferred key points were above the confidence threshold of 0.6. **i** The performance of pose-estimation networks trained with real data deteriorated drastically when they were put to work on recordings of the same species obtained under different conditions, demonstrating their reliance on specimen-independent features. This domain-specificity is most evident in the handheld recordings, where camera orientation, lighting, and background change continuously: networks trained on real platform data performed poorly, but networks trained solely on synthetic data approach benchmark performance with the addition of just five hand annotated example images (i, refine 25%), indicating a stronger specimen-specific understanding (see Supplementary Videos 5–7). Source data are provided as a Source data file.

location of 46 key points distributed along the body (Fig. 5a–c, see "Methods" and Supplementary Fig. 1 for further details). Dataset generation took about three hours on a consumer-grade laptop (6 Core 4 GHz CPU, 16 GB RAM, RTX 2070 Super).

The synthetic dataset was then used to train a DeepLabCut[20] (DLC) pose estimator with a ResNet101 backbone. We emphasise that other excellent markerless pose estimators, such as SLEAP[16] or

DeepPoseKit[17], exist, and merely use DLC by way of example. The best tool will likely depend on the specific use-case. To test pose-estimation performance, two datasets of walking sunny stick insects were curated as test cases (Fig. 5). One dataset, denoted "platform", represents a typical controlled case, where lighting and image background are constant, and only camera perspective varies: *S. inexpectata* were recorded walking across an evenly lit, tiltable platform, at 55 fps with

five synchronised machine vision cameras (Fig. 5d, e). From these recordings, all 49 key points were hand-annotated in each of 805 frames (see Supplementary Table 10 for split details). The second dataset, denoted "handheld", consists of 200 hand-annotated frames from ten handheld videos (20 frames per video), recorded with a cell phone at 25 fps (Fig. 5h). This dataset represents an uncontrolled case with variable recording conditions, and includes motion blur, perspective and magnification changes, out-of-focus frames, and frequent partial occlusions.

A DLC network trained on frames from all camera perspectives achieved a benchmark mean relative error on platform data of 10.9% across all camera views (Fig. 5g, see Eq. (4)). In remarkable contrast, networks trained exclusively on data from a single camera perspective produced a mean relative error of up to 86.4% for frames from unseen perspectives. This poor performance partially reflects domain-sensitivity, characteristic of many transfer learning approaches: the networks fail to generalise, because they have only been exposed to a small inference-specific dataset with limited variation.

A DLC network trained exclusively with synthetic data seemingly competes with the benchmark performance out-of-the-box: it achieved a mean relative error of 5.89% across all platform camera views (Fig. 5g). However, this low error is deceiving, as the network assigned low confidence scores to more than 50% of key points, which are thus excluded from the error estimate. However, the network provides an excellent starting point for refinement: the provision of a mere ten hand-annotated frames per camera orientation suffices to estimate key points with a mean relative error of 8.14%—better than benchmark performance.

The limited ability of networks trained on real data to generalise becomes even more apparent when they are put to work on recordings of the same species, recorded under different conditions. The pose-estimation network trained on the full platform dataset achieved a mean relative error of 77.18% on the handheld dataset (Fig. 5g). Key points were frequently placed more than two body lengths away from the specimen, demonstrating that key point detection strongly relies on recording-specific latent features; the volume and variability of the supplied training data was insufficient to embed a general specimen-specific understanding. In sharp contrast, the network trained solely on synthetic data achieved a mean relative error of 6.25% on handheld recordings—more than an order of magnitude smaller. Refinement with five randomly sampled frames from each video resulted in a mean relative error of 5.03%, close to the benchmark performance of 3.55%, achieved by a network trained on the full handheld dataset (Fig. 5g, i). An extensive quantitative comparisons of the performance across inference cases is provided in the supplementary information (see Supplementary Table 10 as well as Supplementary Table 11).

On the basis of the above, we conclude that the large sample size and variability afforded by synthetic data can meaningfully increase the domain-invariance and robustness of pose-estimation networks, and thus substantially reduce the required user effort: better or near-benchmark performance was achieved with 4-fold and 16-fold fewer hand-annotated samples in the handheld and the platform case, respectively (Fig. 5e–i, see also Supplementary Videos 5 and 6).

**Semantic segmentation.** We have demonstrated that synthetic data generated by *replicAnt* can substantially reduce the hand-annotation required to power accurate detection, tracking and pose-estimation, or even render it obsolete. Next, we show that it can enable inference in applications for which hand-annotation is so onerous that it is unlikely to be performed at the required scale for all but the most common objects: semantic segmentation, a computer vision task involving pixel-level classification (Fig. 6).

*replicAnt* was used to generate a digital population of 20 leaf-footed bugs (*Leptoglossus zonatus*, Dallas, 1852), based on a 3D model of an adult specimen produced with *scAnt*[51] (Fig. 6a, b). Two synthetic datasets were generated, each encompassing 10,000 images with 1024 × 1024 px resolution (Fig. 6c): one dataset used *replicAnt*'s default parameters; for the other, *replicAnt*'s asset library was supplemented with various plant models from the Quixel Asset library (Epic Games, Inc.), in order to simulate the image content of typical field macro-photographs. Data generation took between 6 and 10 h on a consumer-grade laptop (6 Core 4 GHz CPU, 16 GB RAM, RTX 2070 Super) for the default and plant case, respectively.

Images from both datasets were combined to form a single dataset, used to train Mask-R-CNN[6], UperNet + SWIN transformer[63], and PSPNet[64] networks (see methods and Supplementary Table 12 for details). Producing large validation datasets is infeasible for highly specific semantic segmentation tasks—the very reason why synthetic data is so helpful in these applications. To provide an indicative performance metric, we extracted network masking accuracy for a small number of hand-annotated image examples via the Average Class-wise Recall (ACR. See Eq. (5)).

All trained networks were able to identify the majority of specimen pixels, and segmented few background pixels (Fig. 6d, e). Remarkably, PSPNet, the oldest tested architecture, produced the most accurate segmentations at both high and low magnification, and even in the presence of partially occluded or out-of-focus bodyparts: it achieved an ACR of 94.03% (Fig. 6d, e). Overall mask quality was lowest for Mask-R-CNN, which struggled with high-aspect ratio appendages at higher magnification and with images with higher background noise (ACR of 82.3%). This problem may in part be specific to the particular implementation of Mask-R-CNN, which was trained using lower resolution segmentation polygons instead of per-pixel segmentation mask encoding (see "Methods" for details). Mask-R-CNN does however additionally produce instance segmentations, useful where images contain more than one individual. To utilise this feature, a Mask-R-CNN network was trained on 10,000 synthetically generated images of leafcutter ants (Figs. 2a–c and 6f; see "Methods" for training details), and used to run inference on photographs of foraging *Atta*. The produced masks were of high quality, and contained few false positives (Fig. 6g).

## Discussion

Deep learning-based computer vision methods promise to fundamentally alter what is possible in animal behavioural research[12–18]. A key remaining bottleneck is the "data-hunger" of supervised learning techniques: annotated datasets of the size and variability required to achieve robust, domain-invariant performance are rarely available, and in any case time-intensive to produce[44,65]. One strategy to overcome this limitation is to produce annotated data synthetically, using sufficiently realistic computer simulations[30–32,39,40,42,44,47–50,66]. In order to facilitate this process, we developed *replicAnt*: a synthetic data generator built in Unreal Engine 5 and Python. *replicAnt* is designed to run on consumer grade hardware, and can generate around 1000 annotated images per hour. We provide extensive documentation, parser scripts for popular deep learning frameworks, pre-trained networks for all listed applications, benchmark datasets, additional software to aid automated detection-based buffer-and-recover tracking and 2D pose-estimation[56], and a growing library of ready-to-use 3D animal models.

The utility of *replicAnt* was demonstrated by using it to train deep neural networks for stereotypical tasks in animal behavioural research. In multi-animal detection, tracking, and semantic segmentation, networks trained exclusively with synthetic data achieved a performance sufficient to remove the need for hand-annotation altogether. In markerless pose-estimation, pre-training networks on synthetic data increased the subject-specific understanding of the networks, so enabling a reduction of the amount of hand-annotation required to achieve benchmark performance by more than one order of magnitude (Fig. 5g). The resulting reduction in time costs enables broad

comparative studies—of key biological importance, but currently absent from the literature[67]. We hope that open sharing of 3D models, test data and trained networks will decrease the need for case-specific refinement, and eventually lead to powerful "generalist" networks.

Ample of opportunity for expansion of *replicAnt* exists. For example, the combination of depth passes and camera intrinsics and extrinsics can in principle be used to train networks to infer 3D locations directly from a single 2D image[36,40,68]; informing posture variation

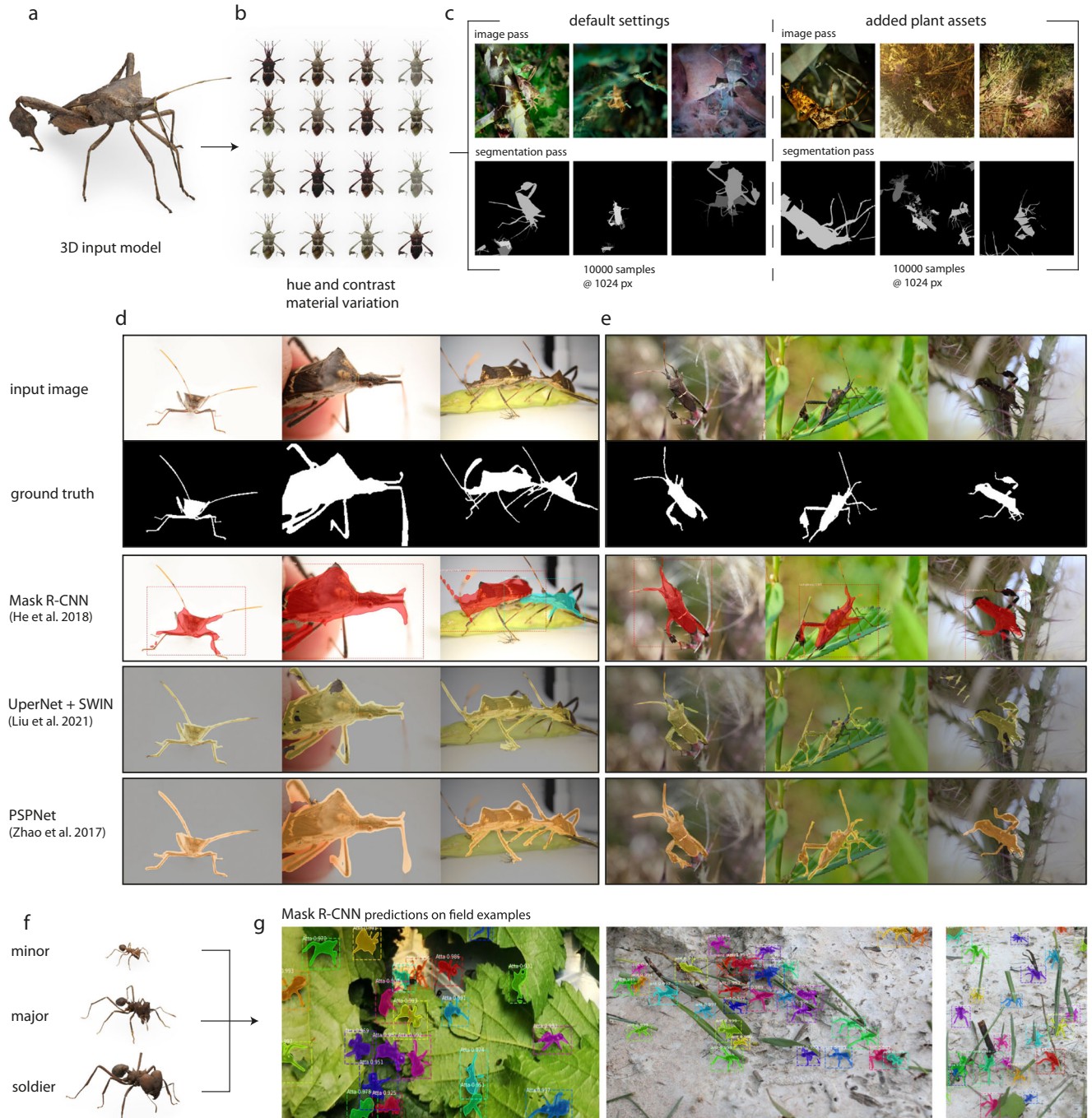

**Fig. 6 | Performance of semantic- and instance segmentation networks trained exclusively on synthetic data. a, b** A population of *Leptoglossus zonatus* leaf-footed bugs was simulated from a 3D model of an adult, produced with the open photogrammetry platform *scAnt*[51]. **c** This population was used to generate two datasets, each encompassing 10,000 images: one with *replicAnt*'s default settings; the other with additional scatterers which placed 3D assets of plants to simulate the image content of typical macro photographs. Images show examples of the image (top row) and ID (segmentation) passes (bottom row), respectively. **d, e** Three deep convolutional semantic segmentation networks—Mask-R-CNN[6], UperNet + SWIN Transformer[63], and PSPNet[64]—were trained on the synthetic data, and their performance was assessed on a small number of hand-annotated **d** laboratory and

**e** field macro-photographs. The highest Average Class-wise Recall (ACR) of 94.03% was achieved by the PSPNet architecture with a ResNet101 back-bone. The Uper-Net + SWIN Transformer network achieved an intermediate overall performance, reduced by fragmented masks and false positives. Segmentations produced by the Mask-R-CNN network had the lowest ACR of 82.3%, but as Mask-R-CNN additionally performs instance segmentation, it is attractive for images that contain multiple individuals. **f, g** To illustrate this feature, a Mask-R-CNN network was trained on 10,000 synthetic images of *Atta vollenweideri* ant workers (Fig. 2), and used to segment **g** crowded laboratory (left) and field photographs (right) of foraging *Atta* leaf-cutter ants.

within *replicAnt* with 3D kinematics data from live animals[30,49] may yield networks which can infer the location of occluded key points with reasonable accuracy; and further annotation, for example automatically labelling minute morphological species differences or body size, can readily be appended. Recent advancements in style- and domain transfer could be combined with data produced by *replicAnt* to produce even stronger generalist[30,31,66], or application specific networks[47–49]. The domain-gap may be narrowed further by introduction of novel pre-trained networks, such as Segment Anything[69] and DINO(v2)[33], as feature-extraction backbones.

Although in principle applicable to any subject model, the experiments and results presented in this study focused primarily on terrestrial arthropods. Further functionality, such as a quadruped armature and models[46,49], animation blueprints, and use of Unreal Engine's soft body and fur simulation capabilities for realistic renderings are planned, to further decrease the entry barrier, and emphasise accessibility and modularity.

*replicAnt* provides a fertile testing ground to further our fundamental understanding of supervised learning. As an illustrative example, *replicAnt* can in principle generate arbitrarily large datasets. Not all images are created equal, however, and the control over environmental variability, combined with the ability for "mixed training", provides an excellent opportunity to probe which image elements are most effective for accelerated network fitting. Ultimately, it is our hope that *replicAnt* represents a significant step towards porting machine-learning based computer vision tools to the field.

## Methods
The generation of synthetic datasets with *replicAnt* can be divided into three steps: (i) 3D Subject Model creation and preparation; (ii) set-up of the generator in Unreal Engine; and (iii) parsing generator outputs into common machine learning data formats to train deep neural networks of choice. In the following sections, we first outline the general structure of the data generation process; application specific details are provided at the end. A glossary is provided in Supplementary Note 1, and detailed documentation and interactive Jupyter notebooks for data parsing and benchmarking are available on GitHub (https://github.com/evo-biomech/replicAnt).

### 3D subject models
In principle, 3D models used in *replicAnt* can come from any source. However, model fidelity is a primary determinant of the simulation-reality-gap, and thus influences the performance that can be achieved. Practically, the required model fidelity depends on the desired application. For example, in applications which typically involve low resolution footage, such as multi-animal tracking (Figs. 2–4), lower fidelity 3D models may suffice; networks used for high-resolution semantic segmentation (Fig. 6) or pose-estimation (Fig. 5), in turn, require models with higher fidelity. We used the open-source photogrammetry platform *scAnt*[55] to produce high fidelity models, and Blender v3.1 to sculpt lower fidelity models. Irrespective of model origin, the model mesh may need to be cleaned, retopologised, and rigged prior to import in the generator. For all models used in this study, this process was completed using Blender v2.92 & 3.1.

**Clean-up.** All unconnected vertices and floating artefacts of the model mesh were deleted, and surfaces cleaned, using Blender's native editing and sculpting tools. Holes were closed by collapsing surrounding vertices to a single point, and/or rebuilding the surrounding topology. The re-connected mesh regions were made seamless by projecting adjacent texture information onto the collapsed or newly created area, respectively; any overlapping vertices and self-intersecting faces were removed (for further details, see ref. 55).

**Retopologising.** To accelerate the data generation process and to allow for larger simulated digital populations, we decreased the mesh resolution of each model to between 1000 and 10,000 vertices, using Blender's native decimate modifier. The number of vertices was chosen such that the overall shape was preserved, but fine topology information, such as hairs and other surface detail were removed; this information was captured by albedo and normal maps instead, generated for the retopologised meshes through texture baking from the original high-resolution input[55].

**Rigging.** In order to enable posture variation, models were rigged—each model was assigned a set of rigid segments referred to as bones, with individual bones connected through joints. The collection of bones and joints defines the model's armature. In principle, users can assign an arbitrary number of virtual bones and joints, each with a specific range of motion. We provide a base armature template that was used throughout this work; it can readily be adapted to animal-specific needs (see Supplementary Fig. 1). The segment deformation associated with joint movement was restricted to proximal parts of each mesh segment using weight painting, an appropriate simplification due to the effectively rigid arthropod exoskeleton.

**Model porting to Unreal Engine.** Curated models, including materials, textures and the armature, were transferred from Blender to Unreal engine 5 via the send-to-unreal Add-on (https://github.com/EpicGames/BlenderTools). Rigged meshes can also be imported directly into Unreal Engine 5, but may then require additional manual editing. Regardless of model origin, process shader, scaling, collisions properties, and animation blueprints need to be assigned to the subject model after import. Examples of configured 3D subject models as well as detailed documentation on the model preparation process are available on GitHub (https://github.com/evo-biomech/replicAnt).

### *replicAnt*
The core of *replicAnt* is a pre-configured Unreal Engine 5 project that includes: (i) example subject models; (ii) project configurations, referred to as levels within Unreal Engine 5; (iii) an asset library—a curated collection of 3D meshes which may populate the generated scenes; (iv) a set of basic materials—image maps that are blended with procedurally generated textures to define the appearance of scattered assets and the ground; and (v) High Dynamic Range panorama images (HDRIs), used as the environment background and to provide additional scene lighting.

These elements form the backbone for the dataset generation process, which can be controlled through a simple user interface; more advanced configuration is possible through editable blueprints, a visual and structured node-based alternative to the use of text-based source code. User-control is facilitated through four main components: (1) The User Interface editor widget (UI); (2) the content browser; (3) the outliner; and (4) the viewport.

The UI is divided into four tabs: General—to define the randomisation schedule of all generator elements, to control the balance between generation speed and output variability, to configure the desired output types and location, and to set the number of unique output samples; Subjects—to select 3D models for inclusion in the digital population, and to specify subject population size and appearance randomisation; Environment—to select HDRIs and to specify the ground mesh resolution; and Debug—to control various stalling periods related to stability, and the randomisation seed. The seed value controls the randomisation routines of all elements and so enables repeatability. For example, identical scenes can be populated with different subjects, so producing datasets of different animals in equal environments.

The Content Browser is akin to a file explorer, and provides access to the full project data structure, all assets and blueprints. Additional

content, such as textures or 3D models, can be imported, and key individual components of the data generator can be modified (for additional documentation, refer to https://docs.unrealengine.com/5.0/). The Outliner displays and provides access to all elements included in the current project configuration, for example the generated subject population, the scene camera, the ground plane, and lights. Users can add, edit, or remove randomisable elements such as asset scatterers, light blocking elements, and decal generators (for details, refer to the respective sections below).

Finally, the viewport provides a 3D preview of the current project state. By selecting "change preview environment" from within the Environment tab of the UI, or by selecting "Test" at the bottom of any of the UI tabs, users can directly assess how the current settings translate into the randomised scene generation process, through either randomising the lighting setup or executing a complete randomisation iteration respectively. The viewport also provides a window into active dataset generation processes: it displays snapshots of output passes, and indicates both the time elapsed and an estimated time to completion.

Dataset generation then proceeds through simulation of scenes, defined by a series of randomisable elements as detailed below. In order to generate datasets with maximal variability in minimal time, the frequency with which scene elements are updated is tied to the time it takes to execute each update. To rationally guide this process, we defined a hierarchy from computationally most to least demanding randomisation components: (i) ground plane, (ii) asset, (iii) subject placement and pose, (iv) material, lighting, camera intrinsics and extrinsics, and image post processing. Elements lower in this hierarchy influence elements upstream, and the frequency with which each element is updated can be user-controlled (Fig. 1). We now briefly explain the basic scene generation at each of these hierarchical element levels.

**Environment.** The environment, which we define as all scene elements aside from subjects, is procedurally built from a number of hierarchically linked modules of surface and material generators, asset scatterers, and a dynamic lighting system. In the first step, a ground plane is generated and tessellated to create a parameterisable surface of up to 5,120,000 triangles. All assets are later placed on this ground, initially created as a 2D surface plane. Height variation is then encoded through a three-channel RGB map. Each channel controls different aspects of the terrain generation process, a design choice which enables independent variation of terrain topology and terrain material. The RGB map is procedurally generated through a set of sub-modules which introduce mesh noise of variable granularity and geometric pattern. In order to maximise variation in terrain topology and material, the generated noise is blended with a curated library of monochromatic displacement maps. The red channel encodes the displacement map for the tessellated ground plane. High values ($\geq 0.5$) indicate a positive, and low values ($\leq 0.5$) a negative shift in height relative to the mean height, respectively; a value of 0.5 thus corresponds to mean ground plane height. The green channel encodes additional noise to blend different materials. The blue channel provides an opportunity for additional user-defined variation; it does not affect default randomisation routines.

**Asset scatterers.** In order to increase the variability of the environment, 3D assets—texture-less photogrammetry models of common objects—are placed on the terrain by a user-defined number of asset scatterers (all models used in this study are from Quixel AB, Epic Games). Each scatterer is assigned a number of assets, one of which is randomly selected per iteration (see "Methods—*replicAnt*"). Each scatterer is also assigned a material at random; assets spawned by the same scatterer thus share the same material, so that no separate texture information is required for each asset mesh. As a result, the project file size remains small, even though the numbers of meshes in the

asset library is large. A number of scatterer presets are provided. The number of scatterers and their individual configuration can be set in the Outliner, which provides further control over key parameters of each scatterer: (i) the set of assets which may be scattered; (ii) the asset size range (all assets are by default of equal size, that is their largest length in X,Y,Z coordinate space is normalised); and (iii) the asset number range, defining the minimum and maximum number of instances of the drawn asset the scatterer may spawn per iteration, respectively.

**Subject placement and posing.** In each randomisation iteration, the generator places subjects drawn from a user-defined "population" at a randomised coordinate. Each subject is subjected to pose variation, which is randomised at two levels: mesh-interacting and mesh-independent posing. Key in this process is an animation blueprint (ABP_InsectBase), which specifies the list of body parts subject to mesh-interacting and mesh-independent posing, respectively, along with the joint types and the joint range of motion. Different subjects within the a population can be assigned individual animation blueprints by generating "child instances" of the original "parent" blueprint.

During mesh-interacting posing, subjects are first scattered and then rotated around their geometric centre, once they are in proximity to the generated ground plane. If a subject intersects with the ground plane or a scattered asset or their randomised location lies below the ground plane, the process is repeated until valid locations are determined for every subject. Each subject is moved along its down-vector until its mesh intersects with any scattered asset, previously spawned subject, or the ground plane. If a subject does not intersect with any mesh along its trace, a new centre-rotation is proposed, and the process repeated until a valid rotation and resulting placement location are found. Possible end-point locations are determined via ray casts; the Unreal Engine 5 internal Full body Inverse Kinematics solver (IKS) is then used to determine permissible joint angles for all body parts assigned in the respective animation blueprint. By default, mesh-interaction posing is performed for all leg segments, i.e. subjects generally stand on other meshes (see Supplementary Fig. 1). By restricting the number of solver iterations, however, some legs can also be left intentionally "mid-air". In such instances, a random point is placed within reach of the most distal part of the IK chain of the respective leg, and the IKS is used to solve for an appropriate joint configuration.

Mesh-independent posing, in turn, controls the pose of other key "bones", such as the head, antennae, or mandibles; the respective joints are each assigned a random angle within the permissible range.

**Materials, lighting, camera and post-processing.** Materials for the terrain, decal and asset layers are generated by independent material generators. Each generator combines randomly generated patterns with a curated library of image textures. Further textures can be added to the respective content directories. The selection of textures, pattern generation, and blending are controlled via the seed value defined in the debug tab of the UI controls.

The environment is then lit using a series of lighting elements: colour-filtered High Dynamic Range Images (HDRIs); a main directional light; randomly placed coloured spotlights, which cast multiple sharp and diffuse shadows, either from scattered assets onto the subjects, or from the subjects themselves onto the environment; light blocking occlusion planes; and volumetric fog, which introduces diffuse lighting and reduced visibility. The array holding the HDRIs can be appended with external images, and the volumetric fog density can be adjusted within the environment tab of the UI. For example, users may wish to match specific lighting conditions of an experimental setup, or simply increase lighting variation further. The number and colour of

spawned spotlights is user-definable (see documentation on GitHub for details).

The combination of ground topology, assets, material and decal layers, subjects and lighting fully defines a specific populated environment. In a last step, a simulated camera is randomly spawned, and oriented such that it points towards a randomly selected subject; this random orientation offset ensures that the subject location relative to the image centre varies across images. Using this camera, an annotated sample image of the randomisation iteration can be captured; each unique camera perspective on an environment constitutes a scene. In addition to camera location and orientation (extrinsics), the generator can also randomise camera intrinsics, such as the simulated sensor size, focal length, aperture, and exposure. The camera randomisation step therefore presents a prime opportunity for computationally inexpensive variation, as variable scenes can be extracted from a single populated environment. In a final step, an array of post-processing filters, providing control over colour temperature and tint, saturation and contrast, vignetting, and various types of grain, are applied. These filters only affect the image render pass (see "Generator outputs"). All camera and post-processing attributes are randomisable (see documentation on GitHub).

## Generator outputs

The generator produces two key outputs: annotation files and image passes. The annotation files comprise a single batch file, containing information of relevance for the whole dataset, and one annotation file per scene, containing image specific annotations. Image passes decode key scene information (see below); custom pass types can be added, or configured passes modified (see GitHub documentation). The desired image pass types, image resolution, compression and output directory are specified in the general tab of the UI.

**Annotation files.** Each dataset is accompanied by a single human-readable batch file, which contains general information: the number of samples, the dataset name, the seed value, and the image pass dimensions. Additionally, it provides references to the IDs internally assigned to each subject within the population, as well as its class, which corresponds to the assigned subject model name, and relative scale. This information is also used by custom data parsers (see below).

For each iteration, the set of image passes is accompanied by a unique human-readable sample file. Sample files document information essential for 3D localisation and pose-estimation applications: the camera intrinsics and extrinsics, including the 3D camera location and rotation, diagonal field of view, and the full view projection matrix. Additionally, sample files provide annotations, such as the coordinates of subject bounding boxes, all 2D key point locations in pixel space, and all 3D key point locations relative to the camera coordinate for every simulated subject. Further custom annotations can be added.

**Image passes.** By default, the generator is configured to write four image pass types per scene: image render, ID, depth, and normal passes. Each pass types encodes different key information.

The image render pass is an RGB colour rendering of the simulated camera view, including image noise, variation in exposure, and depth of field (see "Materials, lighting, camera and post-processing"). The level of compression can be fixed or randomised, and several image formats are available for selection in the general tab of the UI (JPG, PNG, EXR, BMP).

The ID pass encodes subject IDs by assigning all pixels a unique, subject-specific RGB colour value; all non-subject pixels are assigned a value of (0,0,0). Occlusion can thus be determined at pixel-level for the full subject mesh, and for individual key points at the following parser stage. Furthermore, the relative occupancy of the subject mesh within its 2D bounding box can be extracted. Information on occlusion, key point location and bounding box occupancy is essential to exclude

fully occluded subjects or individual key points from neural network fitting where desired. In our illustrative pose-estimation examples (see below), occluded key points were excluded from training; however, users can toggle this option on and off in the respective Jupyter notebook parsers (see GitHub documentation). The ID pass also forms the basis for segmentation maps (Fig. 6c, ID (segmentation) pass) used for run-on encoding, polygon encoding, or per-pixel encoding at the parser stage. ID pass images are always saved in uncompressed PNG format.

The depth pass is a monochromatic render which encodes the depth of each pixel in the scene, relative to the virtual camera plane. In combination with camera intrinsics/extrinsics and the ID pass, depth pass images can in principle be used to train networks for 3D semantic segmentation, size estimation, and depth inference. Depth pass images are always saved in uncompressed PNG format.

The normal pass encodes surface normals, required by some specialised pose-estimation applications[40,68]. Normal passes can be produced both in camera view space and world space. We did not use normal passes in this work, but provide them to illustrate the modularity and extendibility of *replicAnt*. Normal pass images are always saved in uncompressed PNG format.

## Data parsers

The combination of image passes and annotation files constitutes synthetic data which can be used to train a suite of machine-learning based computer vision models. In order to facilitate this process, five groups of data parsers were implemented in form of documented Jupyter notebooks. These parsers translate the generated data into various data formats used by popular deep learning-based computer vision models: (i) YOLO compatible data[4,57]; (ii) three DeepLabCut[20] and SLEAP[16] compatible data parsers; (iii) COCO[6,10] formatted data, including image masks for detection, single and multi-animal pose-estimation, as well as semantic segmentation applications; (iv) MMSegmentation[70] compatible data for segmentation training and benchmarking; and (v) a custom 3D pose-estimation data format which includes camera intrinsics and extrinsics.

The data parsers use a number of common python packages (including *h5py*, *imutils*, *jupyter*, *json5*, *matplotlib*, *opencv-python*, *pandas*, *scikit-image*, *scikit-learn*, *scipy*) to interface with various deep learning frameworks, such as *tensoflow* and *keras*, *darknet*, and *open-mmlab*.

## Applications

The synthetic data generated by *replicAnt* was used in conjunction with a set of machine learning tools, to conduct multi-animal detection, tracking, pose-estimation, and semantic and instance segmentation. For each of these applications, we below describe (1) the curation of hand-annotated benchmark data; (2) the origin of 3D models and the pertinent details of the synthetic data generation process; (3) the network training procedure; and (4) the performance characterisation. By necessity, these applications required to choose specific network architectures for each use case. In choosing specific network architectures over others, we do not wish to imply their superiority; all choices merely serve as illustrative examples, and other options may be better or worse, depending on the specific use case. Our aim is not to determine the best network architecture, but to test the broad applicability and quality of the synthetic data generated by *replicAnt*.

## Detection

**Benchmark data.** Two video datasets were curated to quantify detection performance; one in laboratory and one in field conditions. The laboratory dataset consists of top-down recordings of foraging trails of *Atta vollenweideri* (Forel 1893) leaf-cutter ants. The colony was collected in Uruguay in 2014, and housed in a climate chamber at 25℃ and 60% humidity. As *Atta vollenweideri* neither fall under European

Directive 63/2010/EU, nor are they considered protected species under the Convention on International Trade in Endangered Species (CITES), no specific permits were required. All experiments were designed such that they minimise animal suffering. A recording box was built from clear acrylic, and placed between the colony nest and a box external to the climate chamber, which functioned as feeding site. Bramble leaves were placed in the feeding area prior to each recording session, and ants had access to the recording box at will. The recorded area was 104 mm wide and 200 mm long. An OAK-D camera (OpenCV AI Kit: OAK-D, Luxonis Holding Corporation) was positioned centrally 195 mm above the ground. While keeping the camera position constant, lighting, exposure, and background conditions were varied to create recordings with variable appearance: The "base" case is an evenly lit and well exposed scene with scattered leaf fragments on an otherwise plain white backdrop (Fig. 2). Videos were captured from the OAK-D camera using the accompanying depthai python package (v0.4.0.0). A "bright" and "dark" case are characterised by systematic overexposure or underexposure, respectively, which introduces motion blur, colour-clipped appendages, and extensive flickering and compression artefacts. In a separate well-exposed recording, the clear acrylic backdrop was substituted with a printout of a highly textured forest ground to create a "noisy" case. Last, we decreased the camera distance to 100 mm at constant focal distance, effectively doubling the magnification, and yielding a "close" case, distinguished by out-of-focus workers. All recordings were captured at 25 frames per second (fps).

The field datasets consists of video recordings of *Gnathamitermes* sp. desert termites, filmed close to the nest entrance in the desert of Maricopa County, Arizona, using a Nikon D850 and a Nikkor 18–105 mm lens on a tripod at camera distances between 20 and 40 cm. All video recordings were well exposed, and captured at 23.976 fps.

Each video was trimmed to the first 1000 frames, and contains between 36 and 103 individuals. In total, 5000 and 1000 frames were hand-annotated for the laboratory (Fig. 2e) and field dataset (Fig. 3e), respectively: each visible individual was assigned a constant size bounding box, with a centre coinciding approximately with the geometric centre of the thorax in top-down view. The size of the bounding boxes was chosen such that they were large enough to completely enclose the largest individuals, and was automatically adjusted near the image borders. A custom-written Blender Add-on aided hand-annotation: the Add-on is a semi-automated multi animal tracker, which leverages blender's internal contrast-based motion tracker, but also include track refinement options, and CSV export functionality[55,56]. Comprehensive documentation of this tool and Jupyter notebooks for track visualisation and benchmarking is provided on the *replicAnt* and BlenderMotionExport GitHub[55].

**Synthetic data generation.** Two synthetic datasets, each with a population size of 100, were generated from 3D models of *Atta vollenweideri* leaf-cutter ants. All 3D models were created with the *scAnt* photogrammetry workflow[51]. A "group" population was based on three distinct 3D models of an ant minor (1.1 mg), a media (9.8 mg), and a major (50.1 mg). To approximately simulate the size distribution of *A. vollenweideri* colonies, these models make up 20%, 60%, and 20% of the simulated population, respectively. A 33% within class scale variation, with default hue, contrast, and brightness subject material variation, was used (Fig. 2). A "single" population was generated using the major model only, with 90% scale variation, but equal material variation settings.

A *Gnathamitermes* sp. synthetic dataset was generated from two hand-sculpted models; a worker and a soldier made up 80% and 20% of the simulated population of 100 individuals, respectively with default hue, contrast, and brightness subject material variation (Fig. 3). Both 3D models were created in Blender v3.1, using reference photographs.

Each of the three synthetic datasets contains 10,000 images, rendered at a resolution of 1024 by 1024 px, using the default generator settings as documented in the Generator_example level file (see documentation on GitHub). To assess how the training dataset size affects performance, we trained networks on 100 ("small"), 1000 ("medium"), and 10,000 ("large") subsets of the "group" dataset (see Supplementary Table 2 for dataset sizes and splits). Generating 10,000 samples at the specified resolution took approximately 10 h per dataset on a consumer-grade laptop (6 Core 4 GHz CPU, 16 GB RAM, RTX 2070 Super).

Additionally, five datasets that contain both real and synthetic images were curated. These "mixed" datasets combine image samples from the synthetic "group" dataset with image samples from the real "base" case. The ratio between real and synthetic images across the five datasets varied between 10/1 to 1/100 (see Supplementary Table 3 for dataset sizes and splits).

**Network training.** We used the AlexeyAB darknet implementation of YOLOv4[4] as a detector, because it balances inference quality and speed, and is in widespread use. Each network was trained for 20000 iterations to ensure convergence. The burn-in rate was set to 1000, the learning rate to 10-4, and trained weights were saved every 1000 iterations. As the benchmark data contain recordings with variable subject magnification, we selected a YOLOv4 variant with adjusted anchors, enabling both small and large detections relative to the image size; this variant performed best in preliminary trials.

All training was performed on a computational cluster with compute nodes providing 16 CPU cores, 64 GB of RAM, and a single NVIDIA RTX Quadro 6000 GPU. A comprehensive list of trained networks is provided in the Supplementary Tables.

**Evaluation.** In order to evaluate detection performance, we retrieved the Average Precision (AP) over 13 confidence thresholds, equally spaced between 0.2 to 0.8. A detection was considered correct if the Euclidean distance between its centre and the corresponding ground truth detection was within 5% of the image width; multiple detections of the same object were removed by non-maximum suppression at run-time. To provide a single measure of overall detection performance, we report the mean of the individual APs retrieved for each unseen case (mAP).

We chose a centre-based precision definition over the traditional intersection-over-union (IoU), because different methods were used to assign bounding boxes across hand- and computer annotated data: Synthetically generated bounding boxes represent the smallest rectangle which includes all projected 2D key points in the rendered images; hand-annotated bounding boxes, in turn, are fixed-area, square bounding boxes, as the custom-written centre tracking tool[55] does not report the shape of bounding boxes, and extracts bounding box centres for ease of use instead. Thus, in our application, the IoU is secondary to the proximity of bounding box centres.

The AP was computed as defined in the official scikit learn implementation[71]: it summarises the precision-recall curve with a single weighted mean of the precision at each threshold; the increase in recall from the previous threshold acts as weight:

$$\mathrm{AP} = \sum_n (R_n - R_{n-1}) P_n \tag{1}$$

$$\mathrm{mAP} = \sum_m \frac{\mathrm{AP}_m}{m} \tag{2}$$

Here, $P_n$ and $R_n$ are the precision and recall at the nth threshold. Using decreasing threshold values, the recall $R_{n-1}$ at the first threshold is set to 0; when the threshold is maximal, no detections are returned,

so that the precision $P_O(R_O)$ is equal to unity by definition. This calculation differs from computing the area under the precision-recall curve with the trapezoidal rule, which relies on linear interpolation, and can result in inflated performance estimation[71].

Five-fold cross validation was performed for all trained networks, with an 80/20 data split. Due to the similarity of neighbouring frames from videos recorded with a framerate high compared to the average individual movement speed, validation based on withheld frames alone can artificially inflate accuracy. To avoid this inflation, accuracy was instead assessed by computing the mAP of networks tested only on images outside the original recording domain.

### Tracking

**Benchmark data**. To evaluate multi-animal tracking performance, we used the benchmark datasets curated for the detection experiments—these also provide individual identities across frames due to the annotation with the BlenderMotionExport Add-on[55]. For the laboratory dataset, we used the annotated "base", "dark", "bright", and "noisy" videos, which include between 63 and 103 ant workers. For the field dataset, we use the annotated recording of 49 *Gnathamitermes* sp.

**Synthetic dataset generation and automated tracking**. In principle, sufficiently precise detectors can be used to build simple yet robust and performant detection-based trackers. By retrieving detections for each processed frame, a simple buffer-and-recover tracker can be used to automatically associate detections of adjacent frames to produce coherent tracks and preserve individual identities across frames. The best performing detector models for laboratory and field recordings, trained exclusively on synthetically generated images of *A. vollenweideri* and *Gnathamitermes* sp., respectively, were used for detection-based tracking (see 2). To facilitate tracking, we introduce an open-source Blender Add-on, *OmniTrax*[56]. The best performing YOLO detector models were imported into *OmniTrax*(v0.2.1). *OmniTrax* then uses the loaded network to retrieve input detections for each frame, and to assign detections across frames to specific individuals; tracks are produced automatically and without further user-intervention (for further information, refer to the *OmniTrax* implementation and ref. 56). For all tracking, we used the default tracker and Kalman filter settings, a detector input size of 1088 x 1088, a confidence threshold of 0.5, and non-maximum suppression of overlapping detections of 0.45. Detections with bounding boxes smaller than 20 pixels were automatically excluded from track association.

**Evaluation**. To assess tracker performance, the tracks produced by the detector networks were compared to the hand-annotated ground truth of each frame via the Multiple Object Tracking Accuracy (MOTA)[59]. Calculating the MOTA requires definition of a maximum distance $d_{max}$ between the ground truth detection centres and the inferred tracks beyond which tracks are no longer considered correct. We chose a $d_{max}$ of 50 px at 4k resolution, equivalent to 2% deviation relative to the frame size. The MOTA is then a combined measure of three distinct errors: False Negatives ($FN_t$)—no detection was registered at the location of a ground truth track; False Positives ($FP_t$)—a detection was registered in the absence of a ground truth track within $d_{max}$; ID switches ($IDS_t$)—the identity of an inferred track does not correspond to the identity of the ground truth track previously associated with the inferred track identity. Provided that the identity of an object in the ground truth track at some frame (t) is matched with the identity of an inferred track, we keep track of this correspondence. If, at a later frame $(t + i)$, the ground truth track is associated with a different inferred track, this occurrence is counted as an ID switch. The new identity is then considered "correct" for all subsequent frames, and forms the basis for the identification of any further identity switches. ID switches thus capture "true" identity switches, but also changes in identity due to fragmented tracks that were terminated

early and restarted with a new ID. The MOTA is simply the sum of all $FN_t$, $FP_t$, and $IDS_t$ errors, divided by the sum of all detections $\hat{y}_t$ in the ground truth tracks.

$$\text{MOTA} = 1 - \frac{\sum_t \text{FN}_t + \text{FP}_t + \text{IDS}_t}{\sum_t \hat{y}_t} \tag{3}$$

A maximum MOTA score of unity thus implies that all instances have been detected in every frame, were associated with the correct tracks, and no false positives were produced. Multiple-object tracking evaluation can also yield falsely identified ID switches, $IDS_t$, due to correspondence problems of IDs in close proximity. We keep track of all relevant tracks before and after overlap events, defined as two or more ground truth tracks closer than $d_{max}$, to check whether possible ID switches are simply the result of changes in distances to the respective closest ground truth detection, although identities are retained correctly after the overlap event. Therefore, when assessing overlapping tracks, track identities are only evaluated before and after the event, so that incorrectly identified identity switches are suppressed. If, however, the number of detections is underestimated or overestimated during overlapping events, those occurrences are still counted towards $FN_t$ or $FP_t$, respectively.

### Pose-estimation

**Benchmark data**. Two pose-estimation datasets were procured. Both datasets used first instar *Sungayainexpectata* (Zompro 1996) stick insects as a model species. As *Sungaya inexpectata* do not fall under European Directive 63/2010/EU, nor are they considered protected species under the Convention on International Trade in Endangered Species (CITES), no specific permits are required. All proposed experiments are designed to minimise animal suffering and are unlikely to cause harm. Recordings from an evenly lit platform served as representative for controlled laboratory conditions (Fig. 5e); recordings from a hand-held phone camera (Fig. 5h) served as approximate example for serendipitous recordings in the field.

For the platform experiments, walking *S. inexpectata* were recorded using a calibrated array of five FLIR blackfly colour cameras (Blackfly S USB3, Teledyne FLIR LLC, Wilsonville, Oregon, U.S.), each equipped with 8 mm c-mount lenses (M0828-MPW3 8MM 6MP F2.8-16 C-MOUNT, CBC Co., Ltd., Tokyo, Japan). All videos were recorded via SpinView (v2.7.0.128) at 55 fps, and at the sensors' native resolution of 2048 px by 1536 px. The cameras were synchronised for simultaneous capture from five perspectives (top, front right and left, back right and left), allowing for time-resolved, 3D reconstruction of animal pose via DeepLabCut[20] (DLC) and Anipose[72].

The handheld footage was recorded in landscape orientation with a Huawei P20 (Huawei Technologies Co., Ltd., Shenzhen, China) in stabilised video mode: *S. inexpectata* were recorded walking across cluttered environments (hands, lab benches, PhD desks, etc.), resulting in frequent partial occlusions, magnification changes, and uneven lighting, so creating a more varied pose-estimation dataset. Representative frames were extracted from videos using DeepLabCut (DLC)-internal k-means clustering[20]. Forty-six key points in 805 and 200 frames for the platform and handheld case, respectively, were subsequently hand-annotated using the DLC annotation GUI (see Supplementary Tables 7–9 for additional details regarding dataset splits and composition).

**Synthetic data**. We generated a synthetic dataset of 10,000 images at a resolution of 1500 by 1500 px, based on a 3D model of a first instar *S. inexpectata* specimen, generated with the *scAnt* photogrammetry workflow[51]. Generating 10,000 samples took about three hours on a consumer-grade laptop (6 Core 4 GHz CPU, 16 GB RAM, RTX 2070 Super). We applied 70% scale variation, and enforced hue, brightness,

contrast, and saturation shifts, to generate 10 separate sub-datasets containing 1000 samples each, which were combined to form the full dataset (see Supplementary Table 6 for details).

**Network training.** DeepLabCut[20] version 2.1, built with Tensorflow 2.0 and CUDA 11.2, was used for markerless pose-estimation by way of example. Other excellent choices such as SLEAP[16] or DeepPoseKit[17] exists, and are compatible with the provided data parsers. The best pose-estimation network may depend on the use case, and thus should be chosen by the end-user.

For all experiments, we used a ResNet101 backbone pre-trained on ImageNet[9], with the skeleton configuration and hierarchy as outlined in the base armature (see Supplementary Fig. 1). All networks were trained for 800,000 iterations with a batch size of two; trained weights were saved every 50,000 iterations. These parameters were chosen to prevent excessive overfitting; no further decrease in validation error was observed after 800,000 iterations in preliminary trials. Networks pre-trained on synthetic data were then refined with a small number of real frames for an additional 800,000 iterations (see Supplementary Table 7 for dataset sizes and splits). We used DeepLabCut's default training and augmentation parameters, with image mirroring disabled, to ensure comparability between the trained networks, rather than attempting to tailor network and parameter choice to any singular dataset. All training was performed on a dedicated computational cluster with compute nodes using 16 CPU cores, 32 GB of RAM, and single NVIDIA Quadro RTX 6000 GPU. For a full list of trained networks, refer to the Supplementary Tables 7–9.

**Evaluation.** Pose-estimation performance was quantified as the mean pixel error $\Delta\bar{y}$ and mean relative percentage error $\delta\bar{y}$. The pixel error reported by DLC, $\Delta y$, is the Euclidean distance between the ground truth annotated key point $\hat{y}$ coordinate and the inferred key point coordinate $y$, averaged across all inferred key points with a sufficient confidence value; we chose DLC's default confidence threshold of 0.6. In order to derive a relative error metric, we determined the body length $l$ in pixels as the distance between the most distal point on the head and abdomen for three frames for each ground truth video. Dividing the mean pixel error $\Delta\bar{y}$ by this length proxy yields a resolution-independent measure of pose-estimation performance, expressed as a percentage via:

$$\delta\bar{y} = \frac{100}{n}\sum_{n}\frac{\Delta\bar{y}_n}{l} \tag{4}$$

We performed fivefold cross-validation for all trained networks, with an 80/20 data split between training and withheld data.

### Semantic segmentation

**Benchmark data.** Semantic and instance segmentation is used only rarely in non-human animals, partially due to the laborious process of curating sufficiently large annotated datasets. *replicAnt* can produce pixel-perfect segmentation maps with minimal manual effort. In order to assess the quality of the segmentations inferred by networks trained with these maps, semi-quantitative verification was conducted using a set of macro-photographs of *Leptoglossus zonatus* (Dallas, 1852) and *Leptoglossus phyllopus* (Linnaeus, 1767), provided by Prof. Christine Miller (University of Florida), and Royal Tyler (Bugwood.org; see Fig. 6d–f). For further qualitative assessment of instance segmentation, we used laboratory footage, and field photographs of *Atta vollenweideri* provided by Prof. Flavio Roces (Fig. 6g). More extensive quantitative validation was infeasible, due to the considerable effort involved in hand-annotating larger datasets on a per-pixel basis.

**Synthetic data.** We generated two synthetic datasets from a single 3D scanned *Leptoglossus zonatus* (Dallas, 1852) specimen: one using the default pipeline, and one with additional plant assets, spawned by three dedicated scatterers. The plant assets were taken from the Quixel library and include 20 grass and 11 fern and shrub assets. Two dedicated grass scatterers were configured to spawn between 10,000 and 100,000 instances; the fern and shrub scatterer spawned between 500 to 10,000 instances. A total of 10,000 samples were generated for each sub dataset, leading to a combined dataset comprising 20,000 image render and ID passes. The addition of plant assets was necessary, as many of the macro-photographs also contained truncated plant stems or similar fragments, which networks trained on the default data struggled to distinguish from insect body segments. The ability to simply supplement the asset library underlines one of the main strengths of *replicAnt*: training data can be tailored to specific use cases with minimal effort.

For an additional qualitative demonstration of instance and semantic segmentation, we use the image render and ID passes of the *Atta vollenweideri* "group" dataset also used for detection (see "Synthetic data generation" and Fig. 2a–c).

**Network training.** We trained three different semantic segmentation networks: Mask-R-CNN[6], UperNet with SWIN Transformers[63], and PSPNet[64]. All networks use ResNet101 backbones, and were pre-trained on ImageNet[9]. We used the official Matterport implementation of Mask-R-CNN[6]. For both PSPNet and UperNet training, we use the MMSegmentation[70] implementation, favoured for its versatility and comprehensive architecture support. The generated synthetic data was converted to match the required file format and folder structure conventions, using separate data parsers for the COCO (Common Objects in COntext) and MMSegmentation annotations (see GitHub).

In order to leverage Mask-R-CNN's ability to provide both semantic and instance segmentations, we use the COCO data parser to train a separate Mask-R-CNN model with a ResNet101 backbone, using otherwise identical training parameters and the "group" dataset used also for detection and tracking (see "Results−Detection"). In all cases, the segmentation problem is treated as a binary task: class label of zero or unity are assigned to pixels or polygon areas attributed to the background or any subject, respectively. All networks were trained for a total of 160 epochs, and all training and evaluation of was performed on a desktop workstation with a 14 core CPU, 64 GB of RAM, and a NVIDIA RTX 2080 Ti GPU with 11 GB of VRAM. The full configuration and training schedule is provided with the Semantic And Instance Segmentation Datasets and Trained networks via Zenodo.

**Evaluation.** To provide an indicative quantitative performance metric, we hand-annotated binary masks for the images shown in Fig. 6d, e, and computed the Average Class-wise Recall (ACR):

$$ACR = \frac{100}{c}\sum_{i}^{c}\sum_{j}^{n}\frac{\hat{y}_j - (FP_{i,j} + FN_{i,j})}{\hat{y}_j} \tag{5}$$

Here, FP is the number of false positives−the number of pixels falsely attributed to a class; FN is the number of false negatives−the number of pixels falsely attributed to other classes; and $\hat{y}$ is the total number of pixels in the ground truth mask for each class $c$. The resulting value is multiplied by 100 to provide a percentage score which quantifies the fraction of correctly identified pixels.

### Reporting summary
Further information on research design is available in the Nature Portfolio Reporting Summary linked to this article.

## Data availability
All datasets, both generated and real, additional documentation regarding network fitting, and the best performing networks are hosted via Zenodo and are available from: 3D Models https://zenodo.org/

record/7849059 and DOI: 10.5281/zenodo.7849059; Detection and Tracking Datasets and Trained networks https://zenodo.org/record/7849417 and DOI: 10.5281/zenodo.7849417; Pose-Estimation Datasets and Trained networks https://zenodo.org/record/7849596 and DOI: 10.5281/zenodo.7849596; Semantic And Instance Segmentation Datasets and Trained networks https://zenodo.org/record/7849570 and DOI: 10.5281/zenodo.7849570. Source data are provided with this paper.

## Code availability

All produced code, documentation, and software releases are hosted via GitHub: https://github.com/evo-biomech/replicAnt. The specific version used to generate all synthetic data presented in this study is hosted via Zenodo: https://zenodo.org/record/8378035: DOI: 10.5281/zenodo.8378035.

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

## Acknowledgements

This study received funding from Imperial College's President's PhD Scholarship (to Fabian Plum), and is part of a project that has received funding from the European Research Council (ERC) under the European Union's Horizon 2020 research and innovation programme (Grant agreement No. 851705, to David Labonte). The funders had no role in study design, data collection and analysis, decision to publish, or preparation of the manuscript.

## Author contributions

F.P. and D.L designed the study, analysed and interpreted results, discussed and designed figure content, and wrote the manuscript. F.P. and R.B. designed and implemented the synthetic data generator pipeline, with R.B. contributing major portions of the Unreal Engine code, and F.P. contributing additional Unreal Engine code, as well as all parser, network training, and network evaluation code. F.P. created all 3D subject models, figures, and tables, and performed all network benchmarking. F.P., H.B., and N.I. contributed to the collection and annotation of validation data. All authors have read and agreed upon the contents of this manuscript.

## Competing interests

The authors declare no competing interests.
