## [Peer Review File · Nature Communications]

REVIEWER COMMENTS

Reviewer #1 (Remarks to the Author):

Review:

Summary: The authors provide a framework for training deep learning methods on detecting, segmenting, and reidentifying animals, in particular ants.

Strengths:

- * The paper is well written and clearly lays out the manual steps required (e.g., cleaning up an animal 3D scan before animation) as well as the steps that are automated by this method.
- * The experiments are extensive: on different ant types, in different environments, on different tasks (detection, tracking, segmentation, pose estimation), using different neural network architectures. This diversity shows the great applicability of the proposed method.
- * Sufficient details about the computational environment and user interfaces are given. Providing these details is a strength of the paper.

Weaknesses/required changes:

- * The cleanup and creation of animal models is costly, it would be important for the reader to see intermediate results visually: raw scan, cleaned up mesh, initial rigging, rigging refined after testing, how to set the range of motion, ...
- * More example (videos) of the synthesized examples used for training would be useful. A handful of examples are given for each species but only in low resolution with compression artifacts. Could additional ones be provided in the appendix to get a better sense of the diversity and complexity of the generated footage?
- * The discussion of existing methods for training animal pose and shape estimators is limited, e.g., the following and closely references therein:

A three-dimensional virtual mouse generates synthetic training data for behavioral analysis
L Bolaños, D Xiao, N Ford, J LeDue, P Gupta, C Doebeli, H Hu, H Rhodin, T Murphy
Nature Methods 2021

Deformation-aware Unpaired Image Translation for Pose Estimation on Laboratory Animals
Siyuan Li, Semih Günel, Mirela Ostrek, Pavan Ramdya, Pascal Fua, and Helge Rhodin
CVPR 2020

Suggestions:

- * In the motivation ImageNet and other large scale datasets are mentioned, these are typically used for initializing the neural networks before training on (smaller scale) domain-specific data. Recently, a new generation of pre-trained models has emerged, such as Segment Anything Model (SAM), DINO, and DINOv2... would these be useful for pre-training and alleviating domain gaps? It is worth mentioning these as a future avenue to closing the domain gap.
- * 4.3.2 Image passes contains a missing reference ??
- * I suggest using single-letter variable names in math mode (e.g., E_q), not "error"

Conclusion:

- * The work is of high quality and valuable for the community but needs to be put in context to closely related work.

Reviewer #2 (Remarks to the Author):

The authors describe a pipeline for generating synthetic data for use in various animal tracking deep learning pipelines such as position, identity, and pose tracking and benchmark the performance improvement of this data on various tasks. I think the paper represents an innovative use of game engines to reduce the need for manually labeled data and the authors go beyond prior work in terms of the variety of tasks and data that the method can help generate. I am unfortunately unable to validate the code as I have no access to a Windows 10 machine and don't have the experience to set up a software dev environment in windows. Otherwise I am unable to

find any major issues other than my comments below and therefore if the comments are adequately addressed then I think it is suitable for publication in nature communications.

The title limits the scope of the paper to “unreal engine” but in reality the ideas presented are more general and any 3d game engine could be used. The title should be changed to: “...using a 3d game engine” or similar

The authors make it sound repeatedly like current animal tracking is limited to laboratory settings but this is simply not the case. See for example:

<https://elifesciences.org/articles/47994>

<https://arxiv.org/abs/2103.13282>

<https://besjournals.onlinelibrary.wiley.com/doi/full/10.1111/1365-2656.13904>

The authors tackle many problems that prior work has not, but they should perform their due diligence by thoroughly discussing prior work that uses similar techniques of generating synthetic data to train deep learning algorithms for animal behavior tasks. In particular they should discuss how their work fits into the field and solves problems that these other approaches do not. for example:

<https://www.nature.com/articles/s41592-021-01103-9>

[https://openaccess.thecvf.com/content_ICCV_2019/html/Zuffi_Three-](https://openaccess.thecvf.com/content_ICCV_2019/html/Zuffi_Three-D_Safari_Learning_to_Estimate_Zebra_Pose_Shape_and_Texture_ICCV_2019_paper.html)

[D_Safari_Learning_to_Estimate_Zebra_Pose_Shape_and_Texture_ICCV_2019_paper.html](https://openaccess.thecvf.com/content_ICCV_2019/html/Zuffi_Three-D_Safari_Learning_to_Estimate_Zebra_Pose_Shape_and_Texture_ICCV_2019_paper.html)

<https://arxiv.org/abs/2208.13944>

<https://mshooter.github.io/SyDog/>

<https://arxiv.org/abs/2009.05389>

<https://arxiv.org/abs/2001.08601>

Reviewer #3 (Remarks to the Author):

Plum et al present replicAnt, a software pipeline for synthetic data generation via 3D simulation and rendering aimed at relieving the burden of manual annotation in deep learning-based animal behavior quantification. The authors developed a Unreal-based toolkit for creating synthetic but photorealistic scenes with posed animals with functionality for exporting both the rendered images as well as training targets for neural network-based computer vision tasks including object detection, multi-object tracking, pose estimation and segmentation. They demonstrate that using synthetically generated data, they can train standard computer vision tools for these tasks and that the resulting models are able to generalize favorably to real data collected in both lab and field settings with a variety of imaging setups. The authors also do an exceptional job at thoroughly describing their experiments and evaluation metrics! This is overall a terrific contribution to the field and is likely to have a significant impact owing to the functionality implemented here and the comprehensive evaluations that were conducted.

While this work constitutes a major addition to the field, it should be noted that there have been similar approaches that have been developed that should be referenced to put it in broader context. Additionally, further analyses could be conducted that would strengthen the core claims of the paper.

Major:

1. A discussion comparing this work to related recent works in using synthetic animal data to train computer vision pipelines is necessary, in particular with regard to trade-offs. A notable recent example that should be referenced and appropriately discussed is Bolaños et al 2021 (doi: 10.1038/s41592-021-01103-9) due to the high overlap in domain and target applications.

A discussion, if not explicit comparison, of the strategy of generating more realistic synthetic images to those that aim at narrowing the domain gap algorithmically would be extremely useful for positioning this paper. Some examples: arXiv:1912.08265, arXiv:2003.00080, arXiv:2103.14843

It would also be informative to reference and compare the broader strategy of using photorealistic rendering as compared to purely self-supervised learning for animals (arXiv:2212.07401) or diffusion-based rendering (arXiv:2305.17845).

2. An augmented discussion of the limitations of the framework, in particular as it pertains to mammals and other species, would be especially useful for interested practitioners. For example, how easy is it really to use other models without near-rigid armature/anatomy? Could this be tested easily using publicly available (even pre-rigged) models of mice (as in Bolaños et al)? What about other insects, such as flies? Or other quadrupeds such as in SMAL?

3. Improved sample efficiency is one of the key performance indicators of this approach. While the zero shot examples are impressive, in all likelihood, a researcher will want to fine tune on their own data instead of spending time tweaking the simulation to match their data. Since that's the case, it would be very informative to provide a more systematic evaluation of this by training and evaluating models with varying amounts of real labeled data. The analysis in Fig 2h is helpful, but only done for object detection but not pose/tracking/segmentation. It would also be very informative to see how the performance breaks down across the different real data "cases", where the improvements might vary based on the difficulty of the source data.

Minor:

- Sec 4.3.2 has a missing ref ("see ??")
- Sec 4.9.2 has a missing ref ("Section XX")
- It would be great to provide the datasets generated here as static resources for benchmarking and reproducibility, in addition to the code repository and Unreal projects provided.

Reviewed by Talmo Pereira

RESPONSE TO REVIEWERS' COMMENTS

We thank all referees for their valuable time, and their constructive comments. A point-by-point reply is provided below. One point was raised by all three referees; we thus address it here in order to avoid repetition, and to refer back to it in the point-by-point reply.

All referees suggested additional references which merit discussion. We included all of these references, and some others which we identified as suitable during the search triggered by the comments of the reviewers. Our aim was to appropriately implement the shared suggestion that the work be placed more strongly into context with existing work. In total, we added **14** new references to address this suggestion. In particular, please consider the updated sections in the introduction in lines **L70** to **L91**:

As a result, transfer-learning strategies perform best in well-controlled recording conditions, and additional refinement is required to analyse more variable footage from the gold standard of behavioural studies – field experiments. Although refinement with relatively few hand-annotated samples of the order of a few hundred to a few thousand can enable accurate inference under field conditions[13, 17, 20, 23–26], large appearance deviations from the hand-annotated examples – for example due to changes in weather conditions, recording background, the time of day, or varying camera perspective – typically considerably decrease performance[17, 19, 23, 27]: networks learn latent features specific to the recording environment, rather than a general subject-specific understanding. Some of these generalisation issues can be addressed through data augmentation, i. e. the application of image perturbations with the aim to alter image appearance while retaining its meaning and label[4, 28, 29]. For example, by changing the rotation, scale, hue, and resolution of an image, its contents would still remain identifiable. More sophisticated augmentation strategies, such as style-transfer, can further improve network robustness[30–32]. Alternatively, where large volumes of unlabelled data are available, self-supervised approaches may be employed to learn consistently identifiable features[24, 33]. But these features may then be distinct from case-specific points of interests, in some sense just passing the baton of key-point extraction further down in the analysis pipeline. Currently, even extensive augmentation and unsupervised or self-supervised strategies still pale in their efficacy in comparison to simply using larger and more varied datasets in supervised approaches instead[4, 28, 29].

lines **L108** to **L116**:

More complex approaches have used hand-animated or learned motion priors, or combined low fidelity synthetic data with style- or domain transfer networks to close the simulation-reality gap[27, 30, 31, 48–51]. These approaches however remain labour intense, tied to specific species, possess limited options for annotations, or still require extensive real image datasets in order to generalise to real examples. Comprehensive and generalisable approaches which utilise more realistic animal representations, handle large digital animal populations, can create highly variable environments, and provide options for complex annotation, remain absent.

as well as the additions to the discussion in **L462** to **L480**:

Ample of opportunity for expansion of replicAnt exists. For example, the combination of depth passes and camera intrinsics and extrinsics can in principle be used to train networks to infer 3D locations directly from a single 2D image[36, 40, 69]; informing posture variation within replicAnt with 3D kinematics data from live animals[30, 50], may yield networks which can infer the location of occluded key points with reasonable accuracy; and

further annotation, for example on minute species differences or body size, can readily be appended. Recent advancements in style- and domain transfer could be combined with data produced by replicAnt to produce even stronger generalist[30, 31, 67], or application specific networks[48–50]. The domain-gap may be narrowed further by introduction of novel pre-trained networks, such as Segment Anything[70] and DINO(v2)[33], as feature-extraction backbones.

Although in principle applicable to any subject model, the experiments and results presented in this study focused primarily on terrestrial arthropods. Further functionality, such as a quadruped armature and models[46, 50], animation blueprints, and use of Unreal Engine's soft body and fur simulation capabilities for realistic renderings are planned, to further decrease the entry barrier, and emphasise accessibility and modularity.

Where the discussion of these papers appears short, this is owed to the tight word limit, and not to a lack of appreciation. We thank all referees for pointing out relevant references which we had missed.

Reviewer #1 (Remarks to the Author):

Review:

Summary: The authors provide a framework for training deep learning methods on detecting, segmenting, and reidentifying animals, in particular ants.

Strengths:

- * The paper is well written and clearly lays out the manual steps required (e.g., cleaning up an animal 3D scan before animation) as well as the steps that are automated by this method.
- * The experiments are extensive: on different ant types, in different environments, on different tasks (detection, tracking, segmentation, pose estimation), using different neural network architectures. This diversity shows the great applicability of the proposed method.
- * Sufficient details about the computational environment and user interfaces are given. Providing these details is a strength of the paper.

Weaknesses/required changes:

- * The cleanup and creation of animal models is costly, it would be important for the reader to see intermediate results visually: raw scan, cleaned up mesh, initial rigging, rigging refined after testing, how to set the range of motion, ...

We have added further information on all model processing steps on GitHub (<https://github.com/evo-biomech/replicAnt/tree/main/docs>). We decided to provide this more detailed documentation on GitHub, so the guidelines can be easily updated in response to improvements in methodology, or when new features are added, either from our side or via upgrades to Blender or Unreal. For example, the upcoming release of Unreal Engine 5.3 will introduce rigging and weight painting inside the editor, which removes the need to perform these steps in Blender and makes iterative changes to armatures and animation pipelines easier. Additional details regarding the mesh cleaning process specific to the *scAnt* photogrammetry platform can also be found in the original publication (<https://doi.org/10.7717/peerj.11155>). As a rough guide, mesh clean-up takes an experienced user about one to two hours, and retopologising and rigging take about two hours for the complete insect armature.

- * More example (videos) of the synthesized examples used for training would be useful. A handful of examples are given for each species but only in low resolution with compression artifacts. Could additional ones be provided in the appendix to get a better sense of the diversity and complexity of the generated footage?

We have provided additional full resolution examples for all applications via GitHub (https://github.com/evo-biomech/replicAnt/blob/main/docs/Example_samples.md). In addition, all generated as well as hand-annotated datasets are available on Zenodo (L1103 – L1112):

- 3D Models <https://zenodo.org/record/7849059> DOI : 10.5281/zenodo.7849059
- Detection and Tracking Datasets and Trained networks <https://zenodo.org/record/7849417> DOI : 10.5281/zenodo.7849417
- Pose-Estimation Datasets and Trained networks <https://zenodo.org/record/7849596> DOI : 10.5281/zenodo.7849596
- Semantic And Instance Segmentation Datasets and Trained networks <https://zenodo.org/record/7849570> DOI : 10.5281/zenodo.7849570

* The discussion of existing methods for training animal pose and shape estimators is limited, e.g., the following and closely references therein:

A three-dimensional virtual mouse generates synthetic training data for behavioral analysis
L Bolaños, D Xiao, N Ford, J LeDue, P Gupta, C Doebeli, H Hu, H Rhodin, T Murphy
Nature Methods 2021

Deformation-aware Unpaired Image Translation for Pose Estimation on Laboratory Animals
Siyuan Li, Semih Günel, Mirela Ostrek, Pavan Ramdya, Pascal Fua, and Helge Rhodin
CVPR 2020

We thank the referee for these suggestions, which we have now incorporated alongside additional references in several places in the manuscript (see general reply above).

Suggestions:

* In the motivation ImageNet and other large scale datasets are mentioned, these are typically used for initializing the neural networks before training on (smaller scale) domain-specific data. Recently, a new generation of pre-trained models has emerged, such as Segment Anything Model (SAM), DINO, and DINOv2... would these be useful for pre-training and alleviating domain gaps? It is worth mentioning these as a future avenue to closing the domain gap.

Thank you for these suggestions. We have implemented them in the discussion, see general reply above as well as L471 – L472:

The domain-gap may be narrowed further by introduction of novel pre-trained networks, such as Segment Anything[70] and DINO(v2)[33], as feature-extraction backbones.

* 4.3.2 Image passes contains a missing reference ??

* I suggest using single-letter variable names in math mode (e.g., E.q. 4), not "error"

Both implemented as suggested.

Conclusion:

* The work is of high quality and valuable for the community but needs to be put in context to closely related work.

Reviewer #2 (Remarks to the Author):

The authors describe a pipeline for generating synthetic data for use in various animal tracking deep learning pipelines such as position, identity, and pose tracking and benchmark the performance improvement of this data on various tasks. I think the paper represents an innovative use of game engines to reduce the need for manually labeled data and the authors go beyond prior work in terms of the variety of tasks and data that the

method can help generate. I am unfortunately unable to validate the code as I have no access to a Windows 10 machine and don't have the experience to set up a software dev environment in windows. Otherwise I am unable to find any major issues other than my comments below and therefore if the comments are adequately addressed then I think it is suitable for publication in nature communications.

The title limits the scope of the paper to "unreal engine" but in reality the ideas presented are more general and any 3d game engine could be used. The title should be changed to: "...using a 3d game engine" or similar

We thank the referee for this suggestion. We agree that the general idea could perhaps be implemented in other 3D gaming engines, but we specifically used Unreal Engine, and all tools we developed remain uniquely tied to it. We thus prefer to keep the title as is – the main point is not that one can create synthetic data with a 3D game engine (which has been successfully implemented in various 3D rendering systems in the past), but a particular instantiation of this idea which leverages several powerful features of Unreal Engine.

The authors make it sound repeatedly like current animal tracking is limited to laboratory settings but this is simply not the case. See for example:

<https://elifesciences.org/articles/47994>

<https://arxiv.org/abs/2103.13282>

<https://besjournals.onlinelibrary.wiley.com/doi/full/10.1111/1365-2656.13904>

We apologise that our phrasing led to this impression. Our aim was to highlight the difficulty of generalisation to unseen conditions, such as training a model under laboratory conditions and expecting it to perform similarly under unseen field conditions, without further refinement. We have now changed the phrasing as follows, to clarify that available approaches can and have been applied in field-settings, too (Introduction, **L70** to **L79**):

As a result, transfer-learning strategies perform best in well-controlled recording conditions, and additional refinement is required to analyse more variable footage from the gold standard of behavioural studies – field experiments. Although refinement with relatively few hand-annotated samples of the order of a few hundred to a few thousand can enable accurate inference under field conditions[13, 17, 20, 23–26], large appearance deviations from the hand-annotated examples – for example due to changes in weather conditions, recording background, the time of day, or varying camera perspective – typically considerably decrease performance[17, 19, 23, 27]: networks learn latent features specific to the recording environment, rather than a general subject-specific understanding.

The authors tackle many problems that prior work has not, but they should perform their due diligence by thoroughly discussing prior work that uses similar techniques of generating synthetic data to train deep learning algorithms for animal behavior tasks. In particular they should discuss how their work fits into the field and solves problems that these other approaches do not. for example:

<https://www.nature.com/articles/s41592-021-01103-9>

https://openaccess.thecvf.com/content_ICCV_2019/html/Zuffi_Three-D_Safari_Learning_to_Estimate_Zebra_Pose_Shape_and_Texture_ICCV_2019_paper.html

<https://arxiv.org/abs/2208.13944>

<https://mshooter.github.io/SyDog/>

<https://arxiv.org/abs/2009.05389>

<https://arxiv.org/abs/2001.08601>

We thank the referee for these suggestions. We have now commented on and cited these papers, alongside other relevant references, in several places in the manuscript (see general comment above).

Reviewer #3 (Remarks to the Author):

Plum et al present *replicAnt*, a software pipeline for synthetic data generation via 3D simulation and rendering aimed at relieving the burden of manual annotation in deep learning-based animal behavior quantification. The authors developed a Unreal-based toolkit for creating synthetic but photorealistic scenes with posed animals with functionality for exporting both the rendered images as well as training targets for neural network-based computer vision tasks including object detection, multi-object tracking, pose estimation and segmentation. They demonstrate that using synthetically generated data, they can train standard computer vision tools for these tasks and that the resulting models are able to generalize favorably to real data collected in both lab and field settings with a variety of imaging setups. The authors also do an exceptional job at thoroughly describing their experiments and evaluation metrics! This is overall a terrific contribution to the field and is likely to have a significant impact owing to the functionality implemented here and the comprehensive evaluations that were conducted.

While this work constitutes a major addition to the field, it should be noted that there have been similar approaches that have been developed that should be referenced to put it in broader context. Additionally, further analyses could be conducted that would strengthen the core claims of the paper.

Major:

1. A discussion comparing this work to related recent works in using synthetic animal data to train computer vision pipelines is necessary, in particular with regard to trade-offs. A notable recent example that should be referenced and appropriately discussed is Bolaños et al 2021 (doi: 10.1038/s41592-021-01103-9) due to the high overlap in domain and target applications.

We thank the referee for this suggestion. This work, alongside other references, is now discussed and referred to in several places (see general comment above).

A discussion, if not explicit comparison, of the strategy of generating more realistic synthetic images to those that aim at narrowing the domain gap algorithmically would be extremely useful for positioning this paper. Some examples: arXiv:1912.08265, arXiv:2003.00080, arXiv:2103.14843

We thank the referee for this suggestion. These papers, alongside other references, are now discussed and referred to at several places (see general comment above). We agree that a combination of domain transfer, adversarial learning approaches and synthetically generated data by *replicAnt* will likely yield more competitive results. A more detailed discussion of this idea, however, would indeed require a side-by-side test, which is beyond the scope of this work, if no doubt interesting for future work!

It would also be informative to reference and compare the broader strategy of using photorealistic rendering as compared to purely self-supervised learning for animals (arXiv:2212.07401) or diffusion-based rendering (arXiv:2305.17845).

We thank the referee for this suggestion. These strategies are now discussed as follows (L85 – L91):

Alternatively, where large volumes of unlabelled data are available, self-supervised approaches may be employed to learn consistently identifiable features[24, 33]. But these features may then be distinct from case-specific points of interests, in some sense just passing the baton of key-point extraction further down in the analysis pipeline. Currently, even extensive augmentation and unsupervised or self-supervised strategies still pale in their efficacy in comparison to simply using larger and more varied datasets in supervised approaches instead[4, 28, 29].

2. An augmented discussion of the limitations of the framework, in particular as it pertains to mammals and other species, would be especially useful for interested practitioners. For example, how easy is it really to use other models without near-rigid armature/anatomy? Could this be tested easily using publicly available (even pre-rigged) models of mice (as in Bolaños et al)? What about other insects, such as flies? Or other quadrupeds such as in SMAL?

In light of the tight word limit, we were unfortunately unable to discuss this fair and helpful observation extensively. To acknowledge the point raised by the referee, we instead now more explicitly discuss the restriction of the work to terrestrial arthropods, and discuss the need (but also opportunity) to expand the approach to vertebrates and alternative locomotor modes (discussion, L474 – L480):

Although in principle applicable to any subject model, the experiments and results presented in this study focused primarily on terrestrial arthropods. Further functionality, such as a quadruped armature and models[46, 50], animation blueprints, and use of Unreal Engine's soft body and fur simulation capabilities for realistic renderings are planned, to further decrease the entry barrier, and emphasise accessibility and modularity. Future additions to replicAnt may also address flying and swimming species.

While adding other species is already possible for experienced Unreal users - as *replicAnt* is agnostic to the model origin and animation routines - we are currently working to include other modular base armatures and example models with future releases.

3. Improved sample efficiency is one of the key performance indicators of this approach. While the zero shot examples are impressive, in all likelihood, a researcher will want to fine tune on their own data instead of spending time tweaking the simulation to match their data. Since that's the case, it would be very informative to provide a more systematic evaluation of this by training and evaluating models with varying amounts of real labeled data. The analysis in Fig 2h is helpful, but only done for object detection but not pose/tracking/segmentation. It would also be very informative to see how the performance breaks down across the different real data "cases", where the improvements might vary based on the difficulty of the source data.

We agree with the referee, and have now provided detailed comparisons on data efficiency in four new, dedicated tables for Detection, Tracking, and the two modes of static vs handheld pose estimation in the SI:

- Table A5 : Case-specific detection performance (AP) of YOLOv4 (in Atta)
- Table A6 : Case-specific tracking performance (MOTA) of YOLOv4 networks within OmniTrax (in Atta)
- Table B10 Pose-estimation performance (mean pixel error Δy and mean relative percentage error δy) of DLC with ResNet101 backbone (in Sungaya, platform data).
- Table B11 Pose-estimation performance (mean pixel error Δy and mean relative percentage error δy) of DLC with ResNet101 backbone (in Sungaya, handheld data)

We have not included further tables on data efficiency in semantic segmentation examples, as in this case, all models have already been trained exclusively with synthetic data, without hand-annotated samples used for refinement.

Minor:

- Sec 4.3.2 has a missing ref ("see ??")
- Sec 4.9.2 has a missing ref ("Section XX")

Thank you – both corrected!

- It would be great to provide the datasets generated here as static resources for benchmarking and reproducibility, in addition to the code repository and Unreal projects provided.

All datasets, 3D models, and trained networks are made available on Zenodo. Hopefully they will be of use to others! (See **L1103** – **L1112**, Data Availability.)

- 3D Models <https://zenodo.org/record/7849059> DOI : 10.5281/zenodo.7849059
- Detection and Tracking Datasets and Trained networks
<https://zenodo.org/record/7849417> DOI : 10.5281/zenodo.7849417
- Pose-Estimation Datasets and Trained networks <https://zenodo.org/record/7849596> DOI : 10.5281/zenodo.7849596
- Semantic And Instance Segmentation Datasets and Trained networks
<https://zenodo.org/record/7849570> DOI : 10.5281/zenodo.7849570

Reviewed by Talmo Pereira

REVIEWERS' COMMENTS

Reviewer #1 (Remarks to the Author):

My concerns have been well addressed by the revision, in particular the discussion of additional related work that was also resonated by the other reviewers. I believe with these changes and clarification the paper is ready for publication.

Reviewer #3 (Remarks to the Author):

The authors have now satisfactorily addressed previous critiques through more extensive citations and contextualization of existing work, as well as by providing data on additional experiments to support the advantages of the approach in terms of sample efficiency.

-Talmo Pereira